# CONSTRAINED PHYSICAL-STATISTICS MODELS FOR DYNAMICAL SYSTEM IDENTIFICATION AND PREDICTION

**Jérémie Donà**[*1] **, Marie Déchelle**[*1]**, Marina Levy**[2]**, Patrick Gallinari**[1 3]
[1]Sorbonne Université, CNRS, ISIR, F-75005 Paris, France
[2]Sorbonne Université, CNRS, LOCEAN-IPSL, F-75005 Paris, France
[3]Criteo AI Labs, Paris, France
`firstname.lastname@isir.upmc.fr` except `marina.levy@locean.ipsl.fr`

## ABSTRACT

Modeling dynamical systems combining prior physical knowledge and machine learning (ML) is promising in scientific problems when the underlying processes are not fully understood, e.g. when the dynamics is partially known. A common practice to identify the respective parameters of the physical and ML components is to formulate the problem as supervised learning on observed trajectories. However, this formulation leads to an infinite number of possible decompositions. To solve this ill-posedness, we reformulate the learning problem by introducing an upper bound on the prediction error of a physical-statistical model. This allows us to control the contribution of both the physical and statistical components to the overall prediction. This framework generalizes several existing hybrid schemes proposed in the literature. We provide theoretical guarantees on the well-posedness of our formulation along with a proof of convergence in a simple affine setting. For more complex dynamics, we validate our framework experimentally.

## 1 INTRODUCTION

Dynamical systems prediction and identification find crucial applications ranging from medicine and the study of tumors (Hanahan & Weinberg, 2011; Lu & Fei, 2014) to oceanic and climate forecasting (Oreskes et al., 1994; Caers, 2011). The modeling of such systems traditionally rely on ordinary or partial differential equations (ODE/PDE) (Madec, 2008; Marti et al., 2010), and their resolution via numerical solvers and data assimilation (Ghil & Malanotte-Rizzoli, 1991). In real world applications, two main pitfalls occur: first the dynamics may only be partially known and thus do not fully represent the studied phenomena (Rio & Santoleri, 2018); second, the system state may only be partially observed as in ocean models (Gaultier et al., 2013). Machine learning (ML) has become a complementary approach to traditional physics based models (denoted MB for model based) (Reichstein et al., 2019; Dueben & Bauer, 2018). Both offer advantages: whereas MB approaches generalize and extrapolate better, ML high expressivity approaches benefit from the ongoing growth of available data such as satellite observations, with reduced costs compared to data assimilation.

In that perspective, recent lines of work tackle the learning of hybrid models relying on prior physical knowledge and machine learning (Yin et al., 2021; Mehta et al., 2020). Efficiently learning such decompositions actually means solving two different tasks: system identification, i.e. estimating the parameters of the physical model, and prediction, i.e. recovering the trajectories associated to the dynamics. Both are essential for hybrid MB/ML models of dynamical systems. Whereas prediction aims at robust extrapolation, identification accounts for physical interpretability of the MB/ML model. While solving both problems using model-based formulation admits well-known numerical solutions, for example using the adjoint method (Le Dimet & Talagrand, 1986; Courtier et al., 1994), the combination of physical models and deep learning is still an open area of research. In this context, ML applications mainly focus on the prediction task, at the expense of the system identification: Willard et al. (2020) underlines the lack of generalizability of black-box ML models and their

---

[*]Equal contribution

inability to produce physically sound results. Indeed, Ayed et al. (2020) show that without any prior knowledge, the recovered estimates of a dynamical system states are not physically plausible despite accurate predictions. Moreover, as noted by Yin et al. (2021), learning a linear MB/ML decomposition with the sole supervision on the system trajectories is ill-posed and admits an infinite number of decompositions. Such observations highlight the need to incorporate physically motivated constraints in the learning of hybrid models, e.g. through regularization penalties, and several works propose additional constraints to guide the model towards physical solutions (Jia et al., 2019; Yin et al., 2021; Linial et al., 2021). Finally, to complete prior dynamical knowledge with a data-driven component and ensure interpretability of the decomposition, we work out a principled framework that generalizes previous attempts in the regularization of hybrid models. Our contributions are :

- In section 3.1, we introduce a novel way to recover well-posedness and interpretability in the learning of hybrid MB/ML models via the control of an upper bound. We extend our framework to incorporate auxiliary data when available to handle complex real-world data.

- In section 3.2, we propose a novel alternate-optimization algorithm to learn hybrid models.

- In section 3.3, we provide an analysis of the convergence of the proposed algorithm on a simplified case and experimentally evidence the soundness of our approach on more complex settings of increasing difficulty including challenging real world problems (section 4).

## 2    BACKGROUND AND PROBLEM SETUP

We consider a dynamical system with state at time $t$ denoted $Z_t = Z(t)$. $Z_t$ might be fully or only partially observed: we write $Z_t = (X_t, Y_t)$, where $X_t$ is the observed component and $Y_t$ the unobserved one. The evolution of $Z$ is governed by a differential equation with dynamics :

$$\frac{dZ_t}{dt} = \frac{d}{dt}\begin{pmatrix} X_t \\ Y_t \end{pmatrix} = \begin{pmatrix} f_X(Z_t) \\ f_Y(Z_t) \end{pmatrix} \tag{1}$$

The objective is to predict trajectories of $X$, i.e. to model the evolution of the observable part following $\frac{dX_t}{dt} = f_X(Z_t)$. For simplicity, we omit the index $X$ in $f_X$ and write $f(.) \triangleq f_X(.)$.

**Dynamical Hypothesis**    We assume partial knowledge of the dynamics of the observed $X_t$:

$$\frac{dX_t}{dt} = f(Z_t) = f_k(Z_t) + f_u(Z_t) \tag{2}$$

where $f_k \in \mathcal{H}_k$ is a known operator with unknown parameters $\theta^*$, and $f_u \in \mathcal{H}_u$ is the unknown residual dynamics. $\mathcal{H}_k$ and $\mathcal{H}_u$ denote function spaces, see discussion in appendix B.

**Learning Problem**    Our objective is to approximate $f$ with a function $h$ learned from the observed data. According to eq. (2), we assume $h = h_k + h_u$. $h_k \in \mathcal{H}_k$, i.e. belongs to the same hypothesis space as $f_k$: it has the same parametric form. Its parameters are denoted $\theta_k$. Note that $h_k(., \theta^*) = f_k$. $h_u \in \mathcal{H}_u$ is represented by a free form functional with parameters $\theta_u$, e.g. a neural network. The learning problem is to estimate from data the parameters of $h_k$ so that they match the true physical ones and $h_u$ to approximate at best the unknown dynamics $f$. In this regard, an intuitive training objective is to minimize a distance d between $h = h_k + h_u$ and $f$:

$$\mathrm{d}(h, f) = \mathbb{E}_{Z \sim p_Z}\|h(Z) - f(Z)\|_2, \tag{3}$$

where $p_Z$ is the distribution of the state $Z$ that accounts for varying initial states. Each $Z$ defines a training sample. Minimizing eq. (3) with $h = h_k + h_u$ enables to predict accurate trajectories but may have an infinite number of solutions and $h_u$ may bypass the physical hypothesis $h_k$. Thus, interpretability is not guaranteed. We now develop our proposition to overcome this ill-posedness.

## 3    METHOD

In hybrid modeling, two criteria are essentials: 1. identifiability, i.e. the estimated parameters of $h_k$ should correspond to the true physical ones; 2. prediction power, i.e. the statistical component $h_u$ should complete $h_k$ so that $h = h_k + h_u$ performs accurate prediction over the system states. To

control the contribution of each term $h_k$ and $h_u$, we work upper bounds out of eq. (3) (section 3.1). We then propose to minimize $\mathrm{d}(h, f)$ while constraining the upper bounds, which provide us with a well-posed learning framework (section 3.2). Besides, we show that several previous works that introduced constrained optimization to solve related problems are specific cases of our formulation (Yin et al., 2021; Jia et al., 2019; Linial et al., 2021). Finally, we introduce an alternate optimization algorithm which convergence is shown in section 3.3 for a linear approximation of $f$.

## 3.1 Structural Constraints for Dynamical Systems

To ensure identifiability, we derive regularizations on $h_k$ and $h_u$ flowing from the control of an upper bound of $\mathrm{d}(h, f)$. In particular, to minimize $\mathrm{d}(h_k, f_k)$ would enable us to accurately interpret $h_k$ as the true $f_k$, and $h_u$ as the residual dynamics $f_u$. However, since we do not access the parameters of $f_k$, computing $\mathrm{d}(h_k, f_k)$ is not tractable. We then consider two possible situations. In the first one, the only available information on the physical system is the parametric form of $f_k$ (or equivalently of $h_k$), training thus only relies on observed trajectories (section 3.1.1). In the second one, we consider available auxiliary information about $f_k$ that will be used to minimize the distance between $h_k$ and $f_k$ (section 3.1.2). While the first setting is the more general, the physical prior it relies on is often insufficient to effectively handle real world situations. The second setting makes use of more informative priors and better corresponds to real cases as shown in the experiments (section 4.2).

### 3.1.1 Controlling the ML Component and the MB Hypothesis

We propose a general approach to constrain the learning of hybrid models when one solely access the functional form of $h_k$. In this case, to make $h_k$ accountable in our observed phenomena, a solution is to minimize $\mathrm{d}(h_k, f)$. Following the triangle inequality we link up both errors $\mathrm{d}(h, f)$ and $\mathrm{d}(h_k, f)$ (computations available in appendix C.1):

$$\mathrm{d}(h, f) \leq \mathrm{d}(h, h_k) + \mathrm{d}(h_k, f) = \mathrm{d}(h_u, 0) + \mathrm{d}(h_k, f) \tag{4}$$

We want the physical-statistical model $h = h_k + h_u$ to provide high quality forecasts. Minimizing the sole upper bound does not ensure such aim, as $h_u$ is only penalized through $\mathrm{d}(h_u, 0)$ and is not optimized to contribute to predictions. We thus propose to minimize $\mathrm{d}(h, f)$ while controlling both $\mathrm{d}(h_u, 0)$ and $\mathrm{d}(h_k, f)$. Such a control of the upper bound of eq. (4) amounts to balancing the contribution of the ML and the MB components. This will be formally introduced in section 3.2.

**Link to the Literature** The least action principle on the ML component i.e. constraining $\mathrm{d}(h_u, 0)$ is invoked for a geometric argument in (Yin et al., 2021), and appears as a co-product of the introduction of $\mathrm{d}(h_k, f)$ in eq. (4). Optimizing $\mathrm{d}(h_k, f)$ to match the physical model with observations is investigated in (Forssell & Lindskog, 1997).

The general approach of eq. (4) allows us to perform prediction (via $h$) and system identification (via $h_k$) on simple problems (see section 4.1). The learning of real-world complex dynamics, via data-driven hybrid models, often fails at yielding a physically sound estimation, as illustrated in section 4.2. This suggests that learning complex dynamics requires additional information. In many real-world cases, auxiliary information is available in the form of measurements providing complementary information on $f_k$. Indeed, a common issue in physics is to infer an unobserved variable of interest (in our case $f_k$ parameters $\theta^\star$) from indirect or noisy measurements that we refer to as proxy data. For instance, one can access a physical quantity but only at a coarse resolution, as in (Um et al., 2020; Belbute-Peres et al., 2020) and in the real world example detailed in section 4.2. We show in the next subsection how to incorporate such an information in order to approximate $\mathrm{d}(h_k, f_k)$.

### 3.1.2 Matching the Physical Hypotheses: Introducing Auxiliary Data

We here assume one accesses a proxy of $f_k$, denoted $f_k^{pr} \in \mathcal{H}_k$. Our goal is to adapt our framework to incorporate such auxiliary information, bringing the regularization induced by $f_k^{pr}$ within the scope of the control of an upper bound. This enables us to extend our proposition towards the solving of real world physical problems, still largely unexplored by the ML community. We have:

$$\mathrm{d}(h, f) \leq \mathrm{d}(h, h_k) + \mathrm{d}(h_k, f_k^{pr}) + \Gamma = \mathrm{d}(h_u, 0) + \mathrm{d}(h_k, f_k^{pr}) + \Gamma \tag{5}$$

where $\Gamma$ is a constant of the problem that cannot be optimized (see appendix C.2). In that context, we can benefit from auxiliary information providing us with coarse estimates of $\theta^\star$, denoted $\theta^{pr}$,

such that $f_k^{pr} = h_k(., \theta^{pr}) \approx f_k$. To use the available $\theta^{pr}$ to guide our estimation towards the true parameters $\theta^\star$ of $f_k$, a simple solution is to directly enforce the minimization of $\mathrm{d}(h_k, f_k^{pr})$ in the parameter space by minimizing $\|\theta_k - \theta^{pr}\|_2$, where $\theta_k$ are the parameters of $h_k$. Indeed, because $f_k$ and $f_k^{pr}$ have identical parametric forms (as both belong to the same functional space $\mathcal{H}_k$), minimizing $\|\theta_k - \theta^{pr}\|_2$ will bring $h_k$ closer to $f_k^{pr}$ and thus to $f_k$. As above, we propose to minimize $\mathrm{d}(h, f)$ while controlling both $\mathrm{d}(h_u, 0)$ and $\mathrm{d}(h_k, f_k^{pr})$, as described in section 3.2.

**Link to the Literature**   In (Linial et al., 2021) $f_k^{pr}$ stands for true observations used to constrain a learned latent space, minimizing $\mathrm{d}(h_k, f_k^{pr})$. Jia et al. (2019) uses synthetic data as $f_k^{pr}$ to pre-train their model which amounts to the control an upper bound, see appendix C.3. Finally, this setting finds an extension, when the model $f_k^{pr}$ is a learned model, for example trained using eq. (4), leading to a self-supervision approach described in appendix C.4.

## 3.2   Learning Algorithm and Optimization Problem

From the upper bounds, we first recover the well-posedness of the optimization and derive a theoretical learning scheme (section 3.2.1). We then discuss its practical implementation (section 3.2.2).

### 3.2.1   Well-Posedness and Alternate Optimization Algorithm

**Recovering Well-Posedness**   We reformulate the ill-posed learning of $\min_{h_k, h_u \in \mathcal{H}_k \times \mathcal{H}_u} \mathrm{d}(h, f)$, by instead optimizing $\mathrm{d}(h, f)$ while constraining the upper bounds. Let us define $\mathcal{S}_k$ and $\mathcal{S}_u$ as

$$\mathcal{S}_k = \{\, h_k \in \mathcal{H}_k \mid \ell(h_k) \leq \mu_k \,\} \qquad \mathcal{S}_u = \{\, h_u \in \mathcal{H}_u \mid \mathrm{d}(h_u, 0) \leq \mu_u \,\} \tag{6}$$

where $\mu_k, \mu_u$ are two positive scalars and $\ell(h_k) = \mathrm{d}(h_k, f)$ in the case of section 3.1.1 and $\ell(h_k) = \mathrm{d}(h_k, f_k^{pr})$ in the case of section 3.1.2. Our proposition then amounts to optimizing $\mathrm{d}(h, f)$ over the Minkowski-sum $\mathcal{S}_k + \mathcal{S}_u = \{\, h = h_k + h_u \mid h_k \in \mathcal{S}_k, h_u \in \mathcal{S}_u \,\}$ :

$$\min_{h \in \mathcal{S}_k + \mathcal{S}_u} \mathrm{d}(h, f), \tag{7}$$

This constrained optimization setting enables us to recover the well-posedness of the optimization problem under the relative compactness of the family of function $\mathcal{H}_k$ (proof in appendix D.3).

**Proposition 1** (Well-posedness). *Under the relative compactness of $\mathcal{S}_k$, eq. (7) finds a solution $h$ that writes as $h = h_k + h_u \in \mathcal{S}_k + \mathcal{S}_u$. Moreover, this solution is unique.*

**Alternate Optimization Algorithm**   As the terms in both upper bounds of eqs. (4) and (5) specifically address either $h_k$ or $h_u$, we isolate losses relative to $h_k$ and $h_u$ and alternate projections of $h_k$ on $\mathcal{S}_k$ and $h_u$ on $\mathcal{S}_u$, as described in Algorithm 1. Said otherwise, we learn $h$ by alternately optimizing $h_k$ ($h_u$ being fixed) and $h_u$ ($h_k$ being fixed). In practice, we rely on a dual formulation (see section 3.2.2 and the SGD version of Algorithm 1 in Appendix F).

---

**Algorithm 1** Alternate estimation: General Setting

---

**Result:** Converged $h_k$ and $h_u$
Set $h_u^0 = 0$, $h_k^0 = \min_{h_k \in \mathcal{H}_k} \mathrm{d}(h_k, f)$, $tol \in \mathbb{R}^+$
  **while** $\mathrm{d}(h, f) > tol$ **do**

$$h_k^{n+1} = \arg\min_{h_k \in \mathcal{S}_k} \mathrm{d}(h_k + h_u^n, f); \quad h_u^{n+1} = \arg\min_{h_u \in \mathcal{S}_u} \mathrm{d}(h_k^{n+1} + h_u, f) \tag{8}$$

    $n \leftarrow n + 1$
**end**

---

The convergence of the alternate projections is well studied for the intersection of convex sets or smooth manifolds (von Neumann, 1950; Lewis & Malick, 2008) and has been extended in our setting of Minkowski-sum of convex sets (Lange et al., 2019). Because d as defined in eq. (3) is convex, $\mathcal{S}_u$ and $\mathcal{S}_k$ are convex sets as soon as $\mathcal{H}_k$ and $\mathcal{H}_u$ are convex (Appendix A). Thus, if $\mathrm{d}(., f)$ is strongly convex, eq. (8) finds one and only one solution (Boyd et al., 2004). However, neither the convexity of $\mathcal{H}_u$ nor of $\mathcal{H}_k$ is practically ensured. Nonetheless, we recover the well-posedness of eq. (7) and show the convergence of Algorithm 1 in the simplified case where $h$ is an affine function

of $X_t$ (see section 3.3). For complex PDE where convexity may not hold, we validate our approach experimentally and we evidence in section 4 that this formulation enables us to recover both an interpretable decomposition $h = h_k + h_u$ and improved prediction and identification performances.

### 3.2.2 PRACTICAL OPTIMIZATION

Equation (6) involves the choice of $\mu_k$ and $\mu_u$. In practice, we implement the projection algorithm by descending gradients on the parameters of $h_k$ and $h_u$, with respect to the following losses:

$$\mathcal{L}_k(h_k) = \lambda_h \mathrm{d}(h, f) + \lambda_{h_k} \ell(h_k) \qquad \mathcal{L}_u(h_u) = \lambda_h \mathrm{d}(h, f) + \lambda_{h_u} \mathrm{d}(h_u, 0) \qquad (9)$$

where $\lambda_h, \lambda_{h_k}, \lambda_{h_u}$ are positive real values, dynamically increased/decreased during training. Indeed, $\mathrm{d}(h_u, 0)$ can be interpreted as a *stability loss*, preventing the neural networks to trump the physical component. On the other hand, $\mathrm{d}(h_k, f)$ can be interpreted has an *initialization loss* yield a first estimate of $\theta_k$ explaining the dynamics.

Yet, $f$ being unknown: $\mathrm{d}(h, f)$ is not tractable. To estimate $\mathrm{d}(h, f)$, we rely on the trajectories associated to the dynamics. We minimize the distance between the ODE flows $\phi_h$ and $\phi_f$ defined by $h$ and $f$, $\mathrm{d}_\phi(\phi_h, \phi_f)$, over all initial conditions $X_0$:

$$\mathrm{d}_\phi(\phi_h, \phi_f) = \mathbb{E}_{X_0} \int_{t_0}^{t} \|\phi_h(\tau, X_0) - \phi_f(\tau, X_0)\|_2 d\tau \qquad (10)$$

We have: $\mathrm{d}_\phi(\phi_h, \phi_f) = 0 \Leftrightarrow \mathrm{d}(h, f) = 0$. Definitions of flows for ODE and in depth consideration on these distances are available in appendix A. The gradients of $\mathrm{d}_\phi(\phi_h, \phi_f)$ with respect to the parameters of $h_k$ or $h_u$ can be either estimated analytically using the adjoint method (Chen et al., 2018) or using explicit solvers, e.g. Rk45, and computing the gradients thanks to the backpropagation, see (Onken & Ruthotto, 2020). To compute eq. (10), we rely on a temporal sampling of $X$: our datasets are composed of $n$ sequences of observations of length $N$, $X^i = (X_{t_0}^i, \ldots, X_{t_0+N\Delta t}^i)$, where each sequence $X^i$ follows eq. (2) and corresponds to one initial condition $X_{t_0}^i$. We then sample the space of initial conditions $X_{t_0}^i$ to compute a Monte-Carlo approximation to $\mathrm{d}_\phi(\phi_h, \phi_f)$. Let ODESolve be the function integrating any arbitrary initial state $x_{t_0}$ up to time $t$ with dynamics $h$, so that $x_t = \texttt{ODESolve}(x_{t_0}, h, t)$. The estimate of $\mathrm{d}_\phi(\phi_h, \phi_f)$ then writes as:

$$\mathrm{d}_\phi(\phi_h, \phi_f) \approx \frac{1}{n} \sum_{i=1}^{n} \sum_{j=1}^{N} \left\| \texttt{ODEsolve}(X_{t_0}^i, h, t_j) - X_{t_j}^i \right\|_2$$

Note that the way to compute ODEsolve differs across the experiments (see section 4).

### 3.3 THEORETICAL ANALYSIS FOR A LINEAR APPROXIMATION

We investigate the validity of our proposition when approximating an unknown derivative with an affine function (interpretable first guess approximators). We here consider $h_k$ as a linear function. We do not assume any information on $f$, thus relieving this section from the need of an accurate prior knowledge $f_k$. In this context, we show the convergence of the learning scheme introduced in Algorithm 1 with $\ell = \mathrm{d}(h_k, f)$, demonstrating the validity of our framework in this simplified setting. For more complex cases, for which theoretical analysis cannot be conducted, our framework is validated experimentally in section 4. All proofs of this section are conducted using the distance $\mathrm{d}_\phi$. Let $X^s$ be the unique solution to the initial value problem:

$$\frac{dX_t}{dt} = f(X_t) \quad \text{with} \quad X_{t=0} = X_0 \qquad (11)$$

With $h_k(X) = AX$ and $h_u(X) = D_A$, the affine approximation of $f$ writes as:

$$\frac{dX_t}{dt} = AX_t + D_A \quad \text{with} \quad X_{t=0} = X_0 \qquad (12)$$

where $A \in \mathcal{M}_{p,p}(\mathbb{R})$, $D_A \in \mathbb{R}^p$. We write $X^D$ the solution to eq. (12) and $X^A$ the solution to eq. (12) when $D_A = 0$. The alternate projection algorithm with the distance $\mathrm{d}_\phi$ writes as:

$$\hat{A} = \arg\min_{A} \int_{t_0}^{t} \left\| X^s(\tau) - X^D(\tau) \right\|_2 d\tau + \lambda_A \int_{t_0}^{t} \left\| X^s(\tau) - X^A(\tau) \right\|_2 d\tau \qquad (13)$$

$$\hat{D}_A = \arg\min_{D_A} \int_{t_0}^{t} \left\| X^s(\tau) - X^D(\tau) \right\|_2 d\tau + \lambda_D \|D_A\|_2 \qquad (14)$$

where $\lambda_D, \lambda_A > 0$. As the optimization of eq. (13) is not convex on $A$, the solution existence and uniqueness is not ensured. The well-posedness w.r.t $A$ can be recovered by instead considering a simple discretization scheme, e.g. $X_{t+1} \approx (AX_t + D_A)\Delta t + X_t$ and solving the associated least square regression, which well-posedness is guaranteed, see details in appendix D.2. Such strategy is common practice in system identification. Theoretical considerations on existence and uniqueness of solutions to eqs. (13) and (14) are hard to retrieve. If $A$ is an invertible matrix:

**Proposition 2** (Existence and Uniqueness). *If $\hat{A}$ is invertible, There exists a unique $D_A$, hence a unique $X^D$, solving eq.* (14). *(proof in appendix D.4)*

Finally, formulating Algorithm 1 as a least square problem in an affine setting (see appendix D.5), we prove the convergence of the alternate projection algorithm (appendix D.6) :

**Proposition 3.** *For $\lambda_D$ and $\lambda_A$ sufficiently high, the algorithm that alternates between the estimation of $A$ and the estimation of $D_A$ following eqs.* (13) *and* (14) *converges.*

## 4    EXPERIMENTS

We validate Algorithm 1 on datasets of increasing difficulty (see appendix E), where the system state is either fully or partially observed (resp. section 4.1 and section 4.2). We no longer rely on an affine prior and explicit $h_k$ and $h_u$ for each dataset. Performances are evaluated via standard metrics: MSE (lower is better) and relative Mean Absolute Error (rMAE, lower is better). We assess the relevance of our proposition based on eqs. (4) and (5), against NeuralODE (Chen et al., 2018), Aphynity (Yin et al., 2021) and ablation studies. We denote Ours eq. (4) (resp. Ours eq. (5)) the results when $\ell = \mathrm{d}(h_k, f)$ i.e eq. (4), (resp. $\ell = \mathrm{d}(h_k, f_k^{pr})$ i.e. eq. (5)) When $\mathrm{d}(h_k, f)$ (resp. $\mathrm{d}(h_u, 0)$) is not considered in the optimization, we refer to the results as $\mathrm{d}(h, f) + \mathrm{d}(h_u, 0)$ (resp. $\mathrm{d}(h, f) + \mathrm{d}(h_k, f)$). When $h$ is trained by only minimizing the discrepancy between actual and predicted trajectories the results are denoted «Only $\mathrm{d}(h, f)$». We report between brackets the standard deviation of the metrics over 5 runs and refer to Appendices F and G for training information and additional results.

### 4.1    FULLY OBSERVABLE DYNAMICS

To illustrate the learning scheme induced by eq. (4), we focus on fully observed low dimensional dynamics: a simple example emerging from Newtonian mechanics and a population dynamics model.

**Damped Pendulum (DPL)**    Now a standard benchmark for hybrid models, we consider the motion of a pendulum of length $L$ damped due to viscous friction (Greydanus et al., 2019; Yin et al., 2021). Newtonian mechanics provide an ODE describing the evolution of the angle $x$ of the pendulum:

$$\ddot{x} - g/L \sin(x) + k\dot{x} = 0 \tag{15}$$

We suppose access to observations of the system state $Z = (x, \dot{x})$. We consider as physical motion hypothesis $h_k(x, \theta_k) = \theta_k \sin(x)$. The true pulsation $\theta^* = g/L$ of the pendulum has to be estimated with $\theta_k$. The viscous friction term $k\dot{x}$ remains to be estimated by $h_u$.

**Population Dynamics (LV)**    Lotka-Volterra ODE system models a prey/predator population dynamics describing the growth of the preys ($x$) without predators ($y$), and the extinction of predators without preys, the non linear terms expressing the encounters between both species:

$$\dot{x} = \alpha x - \beta xy, \quad \text{and} \quad \dot{y} = -\gamma y + \delta xy \tag{16}$$

We observe the system state $Z = (x, y)$ and set as prior knowledge: $h_k(x, y) = (\theta_k^1 x, -\theta_k^2 y)$. $\theta^\star = (\alpha, \gamma)$ has to be estimated by $\theta_k = (\theta_k^1, \theta_k^2)$. $h_u$ accounts for the non linear terms $(\beta xy, \delta xy)$.

**Experimental Setting**    For both DPL and LV experiments, we consider the following setting: we sample the space of initial conditions building 100/50/50 trajectories for the train, validation and test sets. The sequences share the same parameters; respectively $(\frac{g}{L}, k)$, for DPL, and $(\alpha, \beta, \gamma, \delta)$ for LV. The parameter $\theta_k$ is set to a neuron (of dimension 1 in the pendulum and 2 for LV) and $h_u$ is a 2-layer MLP. Further experimental details are available in appendices E.1, E.2 and F.

Table 1: Experimental Results for PDL and LV data. The presented metric for parameter evaluation is the rMAE reported in %. Pred. columns report the prediction log MSE on trajectories on test set.

| Model | PDL | | LV | |
|---|---|---|---|---|
| | rMAE$(\theta_k, \theta^\star)$ | Pred. logMSE | rMAE$(\theta_k, \theta^\star)$ | Pred. logMSE |
| Ours eq. (4) | **1.56 (0.009)** | **-13.7 (0.84)** | **7.80 (0.011)** | -9.28 (0.75) |
| Only $\mathrm{d}(h, f)$ | 9.35 (0.04) | -13.3 (0.65) | 24.5 (0.017) | -9.21 (0.91) |
| $\mathrm{d}(h, f) + \mathrm{d}(h_k, f)$ | 1.82 (0.01) | -13.4 (0.56) | 7.91 (0.02) | -9.01 (0.99) |
| $\mathrm{d}(h, f) + \mathrm{d}(h_u, 0)$ | 11.1 (0.03) | -12.9 (0.29) | 9.80 (0.098) | -9.45 (0.55) |
| Aphynity | 6.15 (0.009) | -12.2 (0.13) | 21.1 (0.016) | **-9.89 (0.53)** |
| NeuralODE | – | -10.1 (0.32) | – | -9.11 (1.1) |

**Identification and Prediction Results** Table 1 shows that despite accurate trajectory forecasting, the unconstrained setting «Only $\mathrm{d}(h, f)$» fails at estimating the models parameters, showing the need for regularization for identification. Constraining the norm of the ML component can be insufficient: for LV data, both Aphynity and $\mathrm{d}(h, f) + \mathrm{d}(h_u, 0)$ do not accurately estimate the model parameters. However, the control of $\mathrm{d}(h_k, f)$, following eq. (4), significantly improves the parameter identification for both datasets. Indeed, in the PDL case, $h_k$ and $f$ are (pseudo)-periodic of the same period, hence the gain in the performances. Finally, our proposition based on eq. (4) is able to identify the parameters of DPL and LV equation with a precision of respectively $1.56\%$ and $7.8\%$ beating all consid-

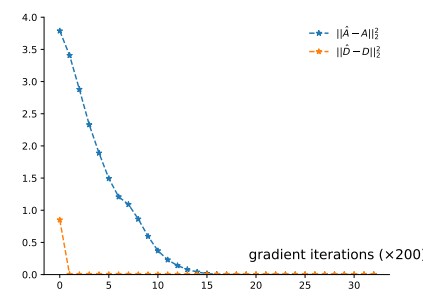

Figure 1: Affine Case : Evolution of the MSE between estimated dynamics $(\hat{A}, \hat{D})$ and the true one $(A, D)$ with the number of gradients steps for linearized DPL.

ered baselines. Regarding prediction performances, in under-constrained settings ( «Only $\mathrm{d}(h, f)$» in Table 1), $h_u$ learns to corrects the inaccurate $h_k$. Table 1 and figs. 4 and 5 (appendix G.1) show that our proposition provides more consistent prediction performances. These experiments confirm that the constraints on $h_k$ and $h_u$ arising from the control of the upper bound of eq. (4) increase interpretability and maintain prediction performances.

**Throwback to the Affine Case** We verify the convergence proved in section 3.3 using the damped pendulum (eq. (15)) linearized in the small oscillations regime (see appendix E.1). Making an affine hypothesis following eq. (12), we apply our alternate projection algorithm and optimize $A$ and $D_A$ alternately using SGD. Figure 1 shows that we are able to accurately estimate $A$ and $D$ using our proposition, recovering both the oscillation pulsation and the damping coefficient.

## 4.2 HIGH DIMENSIONAL DYNAMICS

We now address the learning of transport equations, describing a wide range of physical phenomena such as chemical concentration, fluid dynamics or material properties. We evaluate the learning setting induced by eq. (4) and (5) on two physical datasets depicting the evolution of the temperature $T$ advected by a time-dependent velocity field $U$ and subject to forcing $S$, following:

$$\frac{\partial T}{\partial t} + \nabla.(TU) = S(U) \tag{17}$$

The system state $Z = (T, U, S)$ is partially observed, we only access $T$. Every quantities, observed or to estimate, are regularly sampled on a spatiotemporal grid: at each timestep $t$, the time varying velocity field $U_t$ writes as $U_t = (u_t, v_t)$ and $u_t, v_t, T_t$ and the forcing term $S_t$ are all of size $64 \times 64$.

**Experimental Setting** We consider as physical prior the advection i.e $h_k(T, \theta_k) = -\nabla.(T\theta_k)$. Thus, $\theta_k$ is time-dependent, as we learn it to approximate $\theta^\star = U$. We identify the velocity field $\theta_k$ from observations of $T$, learning a mapping between $T$ and $U$ parameterized by a neural network $G_\psi$, so that $\theta_k = G_\psi(T_{t-l}, ..., T_t) \approx U_t$, which is common practice in oceanography (Béréziat & Herlin, 2015). $G_\psi$ is optimized following eq. (9). $S$ remains to be learned by $h_u$. $h_k$ implements

Table 2: Results for Adv+S and Natl data. We report the MSE ($\times$ 100) on the predicted observations $T$, the velocity fields $U$ and the source term $S$ over 6 time steps on test set.

| Models | Adv+S | | | Natl | | |
|---|---|---|---|---|---|---|
| | $T$ | $U$ | $S$ | $T$ | $U$ | $S$ |
| Ours eq. (4) | **0.74 (0.05)** | **1.99 (0.13)** | **0.17 (0.01)** | 8.27 (0.06) | 11.72 (0.07) | 6.01 (0.08) |
| Ours eq. (5) | – | – | – | **6.86 (0.12)** | **6.81 (0.07)** | **4.35 (0.11)** |
| Aphynity | 0.85 (0.35) | 3.07 (0.74) | 0.18 (0.05) | 8.18 (0.16) | 11.75 (0.49) | 6.02 (0.02) |
| NeuralODE | 1.35 (0.02) | – | – | 8.83 (0.98) | – | – |

a differentiable semi-Lagrangian scheme (Jaderberg et al., 2015) (see appendix E.3) and $h_u$ is a ResNet. $G_\psi$ is a UNet. Training details and a schema of our model are to be found in appendix F.

**Synthetic Advection and Source (Adv+S)** To test the applicability of the learning setting induced by eq. (4) on partially observed settings, we first study a synthetic setting (denoted Adv+S) of eq. (17) by generating velocity fields $U$, simulated following (Boffetta et al., 2001) and adding a source term $S$ inspired by (Frankignoul, 1985). The simulation details are given in appendix E.3.

**Real Ocean Dynamics (Natl)** We consider a dataset emulating real world observations of the North ATLantic ocean (denoted Natl) (Ajayi et al., 2019). Modeling the evolution of $T$ in Natl is challenging as its dynamics is chaotic and highly non-linear. This simulation is representative of the complexity encountered in real world data. The principled approach of eq. (4) is insufficient here and one must resort to additional physical information. We illustrate section 4.2 and make use of auxiliary data: satellite observations provide a coarse estimate of surface velocity fields (appendix E). The goal is to refine the approximated velocity fields to fit the ocean dynamics. We proceed as described in eq. (5) and enforce $d(h_k, f_k^{pr})$ supervising $G_\psi$ with the proxy data (appendix E.3).

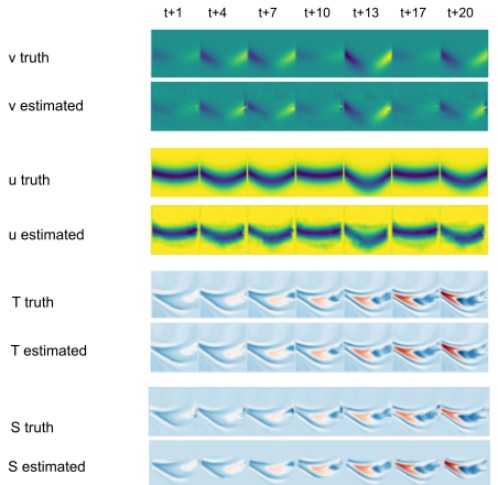

Figure 2: *Best viewed in color*. Estimations of $S, T$ and $U = (u, v)$ on Adv+S. Prediction ranges from 1 to 20 half-days.

**Identification and Prediction Results** Table 2 indicates that for Adv+S dataset, we estimate accurately the unobserved velocity fields. Qualitatively, Figure 2 shows that controlling our proposed upper bound eq. (4) facilitates the recovery of truthful velocity fields $U$ along with an accurate prediction of $T$. For the highly complex Natl, Table 2 shows that the introduction of auxiliary data following the formulation in eq. (5) significantly helps identification, as the dynamics is too complex to be able to recover physically interpretable velocity fields using the bound of eq. (4).

Regarding prediction performances on the Adv+S data, Table 2 shows that thanks to our truthful estimates of $U$, our model provides more precise prediction than NODE and Aphynity. For real world data, thanks to the proxy data our model recovers better velocity fields terms while providing a better estimate for $T$. Besides, adding prior knowledge in the prediction systems improves prediction performances: appendix G shows that NODE minimizes $d(h, f)$ by predicting average and blurred frames. This shows the need for regularization when learning on structured physical data.

**Ablation Study** We present in Table 3 an ablation study on the Adv+S dataset evidencing the influence of our learning choices on the resolution of both identification and prediction tasks (see appendix G for detailed results). "Joint" rows of Table 3 indicate that the learning of $h_u$ and $h_k$ is done simultaneously. Table 3 shows that the sole optimization of $d(h, f)$ fails at estimating physically sounded $U$. This evidences the ill-posedness in such unconstrained optimization. Table 3 indicates that all introduced regularizations improve the recovery of $U$ w.r.t. the «Only $d(h, f)$»

baseline, while adding $\mathrm{d}(h_u, 0)$ significantly improves both prediction performances and velocity fields estimation. We highlight that the alternate optimization performs better compared to optimizing jointly all parameters of $h_k$ and $h_u$. Notably, our proposition to optimize $h_k$ and $h_u$ alternately beats all baselines on both $T$ prediction and $U$ identification (Table 3, Joint rows). Finally, jointly trained models fail at estimating $U$ in Table 3, forcing $h_u$ to capture the whole dynamics.

Table 3: Ablation Study on Adv+S. We report the MSE ($\times$ 100) on the predicted observations $T$, the velocity fields $U$ and the source term $S$ over 6 time steps. "Joint" rows refer to the simultaneous optim. of $h_k$ and $h_u$.

| Training | Models | $T$ | $U$ | $S$ |
|---|---|---|---|---|
| | Ours ($U$ known) | 0.52 | n/a | 0.19 |
| Alternate | Ours eq. (4) | **0.74 (0.05)** | **1.99 (0.13)** | **0.17 (0.01)** |
| | Only $\mathrm{d}(h, f)$ | 1.02 (0.16) | 4.08 (0.23) | 0.19 (0.06) |
| | $\mathrm{d}(h, f) + \mathrm{d}(h_k, f)$ | 1.02 (0.09) | 3.66 (0.15) | 0.19 (0.03) |
| | $\mathrm{d}(h, f) + \mathrm{d}(h_u, 0)$ | 0.77 (0.06) | 2.38 (0.17) | 0.19 (0.01) |
| Joint | Ours eq. (4) | 1.44 (0.08) | 3.30 (0.18) | 0.30 (0.03) |
| | Only $\mathrm{d}(h, f)$ | 1.38 (0.19) | 6.96 (0.21) | 0.39 (0.08) |

## 5 RELATED WORK

Grey-box or hybrid modeling, combining ODE/PDE and data based models, has received an increasing focus in the machine learning community (Rico-Martinez et al., 1994; Thompson & Kramer, 1994; Raissi et al., 2020b). Hybrid approaches allow for alleviated computational costs for fluid simulation (Tompson et al., 2017; De Avila Belbute-Peres et al., 2020; Wandel et al., 2021), and show better prediction performances through data specific constraints that preserve physics (Raissi et al., 2020a; Jia et al., 2019). They offer increased interpretability via constraints on convolutional filters (Long et al., 2018; 2019) or on learned residual (Geneva & Zabaras, 2020). Physical knowledge, introduced through ODE/PDE regularization (Psichogios & Ungar, 1992; Bongard & Lipson, 2007; de Bézenac et al., 2018) or Hamiltonian priors (Greydanus et al., 2019; Lee et al., 2021), increases generalization power w.r.t pure ML approaches. Closer to our work, (Mehta et al., 2020; San & Maulik, 2018; Young et al., 2017; Saha et al., 2020) study the learning of a physical model augmented with a statistical component. Yin et al. (2021) tackle the same task, ensuring the uniqueness in the decomposition by constraining the norm of the ML component. We generalize latter approaches and address the well-posedness in the learning of hybrid ML/MB models through additional regularization on the estimated parameters of the physical part. Indeed, to describe natural phenomena relying on hybrid MB/ML models, one major task lies in the estimation of the MB part parameters. This can be done using neural networks (Raissi et al., 2019; Mehta et al., 2020). However, identification tasks being intrinsically ill-posed (Sabatier, 2000), imposing prior knowledge or regularization is necessary to ensure sound estimations (Stewart & Ermon, 2017). Yet, using only prediction as supervision, the recovered parameters are not physically interpretable (de Bézenac et al., 2018; Ayed et al., 2020). To ensure uniqueness of the estimation solution, Ardizzone et al. (2018) use invertible neural networks. Linial et al. (2021); Tait & Damoulas (2020); Saemundsson et al. (2020) combine variational encoding (Kingma & Welling, 2013) and a PDE model, sampling the space of initial conditions and parameters to solve both identification and prediction. However, such methods only deal with low-dimensional dynamics. Besides low dimensional systems, we also show the soundness of our approach on complex high dimensional and partially observed dynamics.

## 6 DISCUSSION

We propose in this work an algorithm to learn hybrid MB/ML models. For interpretability purposes, we impose constraints flowing from an upper bound of the prediction error and derive a learning algorithm in a general setting. We prove its well posedness and its convergence in a linear approximation setting. Empirically, we evidence the soundness of our approach thanks to ablation studies and comparison with recent baselines on several low and high dimensional datasets. This work can see several extensions: considering non uniform 2 or 3-D grid for climate models, further considerations on the investigated upper bounds, or different decomposition hypothesis.

ACKNOWLEDGEMENTS

We would like to thank all members of the MLIA team from the ISIR laboratory of Sorbonne Université for helpful discussions and comments. We acknowledge financial support from ANR AI Chairs program via the DL4CLIM ANR-19-CHIA- 0018-01 project, the LOCUST ANR project (ANR-15-CE23-0027) and the European Union's Horizon 2020 research and innovation program under grant agreement 825619 (AI4EU). The Natl60 data were provided by MEOM research team, from the IGE laboratory from the Université Grenoble Alpes.

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

# A  DISTANCE

## A.1  DISTANCE BETWEEN DYNAMICS

We here give the definition of the distance d. Let $u$ and $v$ be two functions of $\mathcal{L}^2(\mathbb{R}^p, \mathbb{R}^p)$. We consider the distance:

$$\mathrm{d}(u, v) = \mathbb{E}_{X \sim p_X} \|u(X) - v(X)\|_2 \tag{18}$$

Naturally, eq. (18) verifies the triangle inequality, the symmetry and the positiveness. Moreover, in this case, for all functions $f$, $\mathrm{d}(., f)$ is convex. Indeed, for $u$, $v$ two functions, and $\lambda \in [0, 1]$:

$$
\begin{aligned}
d(\lambda u + (1 - \lambda)v, f) &= \mathbb{E}_{X \sim p_X} \|\lambda u(X) + (1 - \lambda)v(X) - f(X)\|_2 \\
&= \mathbb{E}_{X \sim p_X} \|\lambda u(X) - \lambda f(X) - (1 - \lambda)f(X) + (1 - \lambda)v(X)\|_2 \\
&\leq \lambda \mathbb{E}_{X \sim p_X} \|u(X) - f(X)\|_2 + (1 - \lambda) \mathbb{E}_{X \sim p_X} \|v(X) - f(X)\|_2
\end{aligned}
$$

Hence the convexity of $\mathrm{d}(., f)$. This consideration suffices to ensure the convexity of $\mathcal{S}_k$ and $\mathcal{S}_u$ defined in section 3.

## A.2  DISTANCE BETWEEN FLOWS

Consider the ODE with $X(t), X_0 \in \mathbb{R}^p$:

$$\frac{dX(t)}{dt} = f(X(t)), \quad X(t = 0) = X_0 \tag{19}$$

Equation (19) admits a unique solution as soon as $f$ is Lipschitz. We note $X^\star$ this solution. Then, we can defined the flow $\phi_f$ of such ODE as :

$$
\begin{aligned}
[0,T] \times \mathbb{R}^p &\to \mathbb{R}^p \\
t, \quad X_0 &\to \phi_f(t, X_0) = X^\star(t)
\end{aligned} \tag{20}
$$

With the definition of eq. (20), we can define the distance between two flows of ODE as:

$$d_\phi(\phi_u, \phi_f) = \mathbb{E}_{X_0 \sim p_{X_0}} \int_{t_0}^{\tau} \|\phi_u(t, X_0) - \phi_f(t, X_0)\| \, dt \tag{21}$$

$d_\phi$ is positive and symmetric. Let $\phi_u$, $\phi_v$ be two flows, we have the triangle inequality:

$$
\begin{aligned}
d_\phi(\phi_u, \phi_f) &= \mathbb{E}_{X_0 \sim p_{X_0}} \int_{t_0}^{\tau} \|\phi_u(t, X_0) - \phi_f(t, X_0)\| \, dt \\
&= \mathbb{E}_{X_0 \sim p_{X_0}} \int_{t_0}^{\tau} \|\phi_u(t, X_0) - \phi_v(t, X_0) + \phi_v(t, X_0) + \phi_f(t, X_0)\| \, dt \\
&\leq \mathbb{E}_{X_0 \sim p_{X_0}} \int_{t_0}^{\tau} \|\phi_u(t, X_0) - \phi_v(t, X_0)\| + \|\phi_v(t, X_0) + \phi_f(t, X_0)\| \, dt \\
&\leq d_\phi(\phi_v, \phi_v) + d_\phi(\phi_v, \phi_f)
\end{aligned}
$$

Let $\phi_f$ be fixed, we also have the convexity of $d_\phi(., \phi_f)$ with respect to the first argument. Indeed for $\lambda \in [0, 1]$:

$$
\begin{aligned}
\mathrm{d}_\phi(\lambda \phi_u + (1 - \lambda)\phi_v, f) &= \mathbb{E}_{X_0 \sim p_{X_0}} \int_{t_0}^{\tau} \|\lambda \phi_u(t, X_0) + (1 - \lambda)\phi_v - \phi_f(t, X_0)\| \, dt \\
&= \mathbb{E}_{X_0 \sim p_{X_0}} \int_{t_0}^{\tau} \|\lambda \phi_u(t, X_0) + (1 - \lambda)\phi_v - \lambda \phi_f(t, X_0) - (1 - \lambda)\phi_f(t, X_0)\| \, dt \\
&\leq \lambda \, \mathrm{d}_\phi(\phi_u, \phi_v) + (1 - \lambda)\mathrm{d}_\phi(\phi_v, \phi_f)
\end{aligned}
$$

However, in this case the convexity is not ensured with respect to $u$ and $v$ This is the reason why for theoretical investigations, we consider the distance d instead of $\mathrm{d}_\phi$.

Nonetheless, $\mathrm{d}_\phi(\phi_u, \phi_f) = 0 \implies \phi_u = \phi_f \implies u = f$.

## B    REMARK ON ADDITIVE DECOMPOSITION

First, note that in the case of a metric space the decomposition as defined in eq. (2) always exists.

We now detail an intuition for the well-posedness of such decomposition.

Let $\mathcal{H}_k$ be a closed convex subset of functions of an Hilbert space $(E, <, >)$, and $f$ the function we want to approximate with partial knowledge (represented by the space of hypothesis $\mathcal{H}_k$). Then, thanks to Hilbert projection lemma, we have the uniqueness of the minimizer of $\min_{g \in \mathcal{H}_k} \|f - g\|$, i.e it exists one unique $h_k \in \mathcal{H}_k$ such that: $\forall g \in \mathcal{H}_k, \|f - h_k\| \leq \|f - g\|$.

Finally, the additive decomposition hypothesis presents a remarkable advantage in the case of a vector space. Indeed, if $\mathcal{H}_k$ is a (closed) vector space, let $\mathcal{H}_k^{\perp}$ be its supplementary in $E$, then we have the uniqueness in the decomposition: $f = f_{\mathcal{H}_k} + f_{\mathcal{H}_k^{\perp}}$, where $f_{\mathcal{H}_k^{\perp}} \in \mathcal{H}_k^{\perp}$ and $f_{\mathcal{H}_k} \in \mathcal{H}_k$.

The existence and uniqueness flowing directly from the additive decomposition hypothesis, this can explain why such assumption is common when bridging ML and MB hypothesis.

## C    UPPER BOUNDS

### C.1    DERIVATION OF EQUATION (4)

The first upper bound is a simple use of the triangle inequality:

$$
\begin{aligned}
d(h, f) &= d(h, f) + d(h_k, f) - d(h_k, f) \\
&\leq d(h_k, f) + |d(h, f) - d(h_k, f)| \\
&\leq d(h_k, f) + d(h, h_k)
\end{aligned}
$$

### C.2    DERIVATION OF EQUATION (5)

To derive the second upper bound, we assume that $f_k^{pr}$ comes from an overall dynamics $f^{pr}$ obeying the additive decomposition hypothesis of eq. (2) so that $f^{pr}$ and $f_k^{pr}$ verifies: $f^{pr} = f_k^{pr} + f_u^{pr}$. First, with computations similar to eq. (4), we have:

$$
d(h, f) \leq d(h, f^{pr}) + d(f^{pr}, f) \tag{22}
$$

Then:

$$
\begin{aligned}
d(h, f^{pr}) &= d(h, f^{pr}) + d(h_k, f_k^{pr}) - d(h_k, f_k^{pr}) \\
&\leq d(h_k, f_k^{pr}) + |d(h, f^{pr}) - d(h_k, f_k^{pr})| \\
&\leq d(h_k, f_k^{pr}) + |d(h, f^{pr}) - d(h, f_k^{pr}) - d(h, f_k^{pr}) + d(h_k, f_k^{pr})| \\
&\leq d(h_k, f_k^{pr}) + |d(h, f^{pr}) - d(h, f_k^{pr})| + |d(h_k, f_k^{pr}) - d(h, h_k)| \\
&\leq d(h_k, f_k^{pr}) + d(f^{pr}, f_k^{pr}) + d(h, h_k) \tag{23}
\end{aligned}
$$

Combining Equations (22) and (23), we retrieve eq. (5):

$$
d(h, f) \leq d(h_k, f_k^{pr}) + d(h, h_k) + d(f^{pr}, f_k^{pr}) + d(f^{pr}, f) \tag{24}
$$

and we have: $\Gamma = d(f^{pr}, f_k^{pr}) + d(f^{pr}, f)$. $\Gamma$ is a constant of the problem that cannot be optimized.

### C.3    UPPER-BOUND USING AUXILIARY DYNAMICS $f^{pr}$

Let $f^{pr}$ be the dynamics of model data, we can link up the error made by $h$ on true data (following dynamics $f$) and the error made by $h$ on model data (with dynamics $f^{pr}$) via:

$$
d(h, f) \leq d(h, f^{pr}) + d(f^{pr}, f) \tag{25}
$$

Thus a pre-training on auxiliary data of dynamics $f^{pr}$ amounts to control the term $d(h, f^{pr})$ in the upper-bound of eq. (25).

### C.4 SELF-SUPERVISION

Let $h = h_k + h_u$ be the function to learn and $G_\psi$ the recognition network providing an estimate $\hat{\theta}_k^i$ of the parameters from an initial sequence $(X_{t_0}^i, \ldots, X_{t_0+k\Delta t}^i)$. This learning setting corresponds to how velocity fields are learned from consecutive measurements of the tracer fields $T$ in section 4.2.

To compute $\mathrm{d}(h_k, f_k^{pr})$ in the case where $f^{pr} = h^\star$, where $h^\star = h_k^\star + h_u^\star$ is a learned model, we rely on the computed $\theta_k$ associated to $h_k^\star$ (thanks for example to the algorithm of section 3.2 associated to eq. (4)) to generate a synthetic dataset with achievable supervision in the space of the parameters $\theta_k$.

From a real initial sequence $(X_{t_0}^i, \ldots, X_{t_0+k\Delta t}^i)$, we can estimate the unknown parameter $\theta_k^i$ associated to sequence $i$ with the recognition network $G_\psi^\star$ learned with $h^\star$, i.e $\theta_k^i = G_\psi^\star(X_{t_0}^i, \ldots, X_{t_0+k\Delta t}^i)$. Then, integrating from the initial condition $X_{t_0}^i$, we generate a trajectory of known parameters $\theta_k^i$ with dynamics $h^\star$ denoted by: $\tilde{X}^i = (\tilde{X}_{t_0}^i, \ldots, \tilde{X}_{t_n}^i)$. Sampling the space of initial conditions, we obtain a synthetic dataset: $\big((\tilde{X}^1, \theta_k^1), \ldots, (\tilde{X}^m, \theta_k^m)\big)$ enabling us to perform self-supervision for $G_\psi$. Let $\hat{\theta}_k^i$ be the parameters estimated by $G_\psi$ from the simulated $(\tilde{X}_t^i, \ldots, \tilde{X}_{t+k\Delta t}^i)$, we make the following approximation:

$$\mathrm{d}(h_k, f_k) \approx \frac{1}{m} \sum_{i=1}^{m} \left\| \hat{\theta}_k^i - \theta_k^i \right\|_2 \tag{26}$$

## D PROOFS

### D.1 NOTE ON THE CONVEXITY OF $\mathcal{S}_k$ AND $\mathcal{S}_u$

**Convexity of $\mathcal{S}_k$**

*Proof.* Let $u, v \in \mathcal{S}_k$:
$$\mathrm{d}(tu + (1-t)v, f) = \|tu + (1-t)v - f\| = \|tu - tf + (1-t)v - (1-t)f\|$$
$$\leq t\mu_1 + (1-t)\mu_1 = \mu_1$$

Hence the convexity of $\mathcal{S}_k$. $\qquad\square$

**Convexity of $\mathcal{S}_u$**

*Proof.* Let $t \in [0, 1]$ and $u, v \in \mathcal{S}_u$.
$$\mathrm{d}(h_k, h_k + tu + (1-t)v) = \mathrm{d}(0, tu + (1-t)v)$$
$$\leq t\mathrm{d}(u, 0) + (1-t)\mathrm{d}(v, 0)$$
$$\leq \mu_2$$

Hence the convexity of $\mathcal{S}_u$. $\qquad\square$

### D.2 ODE IDENTIFICATION

Consider the following set: $\mathcal{S}_A = \{X(t) \in \mathcal{C}^1([0,T], \mathbb{R}^p) \text{ such that: } \exists A \in \mathcal{M}_{p,p}(\mathbb{R}), X' = AX\}$, where $T > 0$.

$\mathcal{S}_A$ is not a convex set. Consider $u$ and $v$ in $\mathcal{S}_A$, and consider $A^u$ and $A^v$ so that $u'(t) = A^u u(t)$ and $v'(t) = A^v v(t)$. For $\lambda \in ]0, 1[$: we have:
$$[\lambda u + (1-\lambda)v]' = \lambda u' + (1-\lambda)v'$$
$$= \lambda A^u u + (1-\lambda)A^v v$$

In general the last term is not equal to $A^{\lambda u + (1-\lambda)v}(\lambda u + (1-\lambda)v)$, for some matrix $A^{\lambda u + (1-\lambda)v}$. Thus $\mathcal{S}_A$ is not a convex set. However, discretizing the trajectories and employing a simple integration scheme leads to considering the following cost function:
$$\mathcal{L}(A) = \sum_t \left\| \big(X^s(t+1) - (A\Delta t + Id)X^A(t)\big) \right\|_2^2, \tag{27}$$

As a least square regression problem, $\mathcal{L}(A)$ is convex with respect to $A$. A least square regression setting can also be recovered using more complex integration schemes, or several time steps integration.

### D.3 PROOF FOR WELL-POSEDNESS OF EQUATION (7)

We set ourselves in the Hilbert space of squared integrable functions with the canonical scalar product $\left(\mathcal{L}^2(\mathbb{R}^p, \mathbb{R}^p), <, >\right)$. For further consideration on such functional space we refer to (Droniou, 2001).

We assume that $\mathcal{H}_k$ hence $\mathcal{S}_k$ is convex and a relatively compact family of functions.

**Convexity of $\mathcal{S}_k + \mathcal{S}_u$**  Let $\mathcal{S} = \mathcal{S}_k + \mathcal{S}_u = \{f \mid \exists f_k \in \mathcal{S}_k, f_u \in \mathcal{S}_u, f = f_k + f_u\}$.

Let $f, g \in \mathcal{S}$ and $\lambda \in ]0, 1[$:

$$\lambda f + (1 - \lambda)g = \lambda f_k + (1 - \lambda)g_k + \lambda f_u + (1 - \lambda)g_u \in \mathcal{S}_k + \mathcal{S}_u$$

Hence the convexity of $\mathcal{S}$.

CLOSENESS OF $\mathcal{S}_u$  We show that $\mathcal{S}_u$ is a closed set. Indeed, $\mathcal{S}_u = g^{-1}([0, \mu_u])$, where $g(u) = \|u\|$, Because $g$ is 1-Lipschitz (using the triangle inequality), $g$ is continuous. Therefore $\mathcal{S}_u$ is closed set as the inverse image of a closed set by continuous function.

**Sequential Limit**  We now show that $\mathcal{S}$ is a closed set thanks to the sequential characterisation: let $f^n$ a converging sequence of elements of $\mathcal{S}$ and denote $f$ its limit. We prove that $f^n$ converges in $\mathcal{S}$.

Because $\forall n, f_n \in \mathcal{S}$, we have: $f^n = f_k^n + f_u^n$, where $f_u^n \in \mathcal{S}_u$ and $f_k^n \in \mathcal{S}_k$.

Thanks to the relative compactness of $\mathcal{S}_k$, we can extract a converging sub-sequence, of indexes $n_j$, from $f_k^n$ so that $f_k^{n_j} \to f_k \in \mathcal{S}_k$.

Because $f^n \to f$, the sub-sequence $f^{n_j}$ converges: $f^{n_j} \to f$.

By definition, $f_u^{n_j}$ is a sequence of $\mathcal{S}_u$ and we also have that: $f_u^{n_j} = f^{n_j} - f_k^{n_j}$. Because the right member of the equation converges (as a sum of converging functions), the left member of the equation converges i.e. $f_u^{n_j}$ converges.

Since $\mathcal{S}_u$ is a closed set $f_u^{n_j}$ converges in $\mathcal{S}_u$. We write $f_u$ its limit. Therefore, $f_u^{n_j} = f^{n_j} - f_k^{n_j} \to f - f_k = f_u \in \mathcal{S}_u$. Hence, $f = f_u + f_k$ with $f_u \in \mathcal{S}_u$ and $f_k \in \mathcal{S}_k$.

Therefore $\mathcal{S}$ is a closed set.

Finally, we can apply Hilbert projection lemma on the closed convex set $\mathcal{S}$ and retrieve the uniqueness of the minimizer of eq. (7).

**Remark**  The relative compactness of a family of functions is a common assumption in functional analysis. For example, in the study of differential equation Cauchy-Peano theorem provides the existence to the solution of an ODE under the assumption of relative compactness.

Also, Ascoli theorem provides the relative compactness of a family of function $\mathcal{F}$ under the hypothesis of the equi-continuity of $\mathcal{F}$ and the relative compactness of the image space $A(x) = \{f(x) | f \in \mathcal{F}\}$.

### D.4 PROOF OF PROPOSITION 2

We now set ourselves in the Hilbert space $\left(\mathcal{L}^2([0, T], \mathbb{R}^p), <, >\right)$ of squared integrable functions, where $<, >$ is the canonical scalar product of $\mathcal{L}^2([0, T], \mathbb{R}^p)$.

*Proof.* Let $A$ be a given invertible matrix. We consider the following space $\mathcal{S}_D = \{X \in \mathcal{C}^1([0, T], \mathbb{R}^p)$ such that: $\exists D \in \mathbb{R}^p, X' = AX + D$ and $X(t = 0) = X_0\}$, where $T > 0$. We show that $\mathcal{S}_D$ is a closed convex set.

**Convexity** Indeed, let $\lambda \in ]0,1[$ and $u, v \in \mathcal{S}_D$. $\lambda u + (1 - \lambda)v$ is differentiable and:

$$[\lambda u + (1 - \lambda)v)]' = \lambda u' + (1 - \lambda)v' = A(\lambda u + (1 - \lambda)v) + D,$$

Where $D = \lambda D_u + (1 - \lambda)D_v$. Hence $\lambda u + (1 - \lambda)v \in \mathcal{S}_D$.

**Closeness via Affine-Space** To prove the closeness of $\mathcal{S}_D$, we prove that it is an affine space of finite dimension.

Let $g$ the application that to any vector $D \in \mathbb{R}^d$ associate the solution $X^D$.

Let $D_0 \in \mathbb{R}^D$, we show that $g_{D_0} : D \to g(D_0 + D) - g(D_0)$ is a linear application.

Naturally, for $g_{D_0}(0_{\mathbb{R}^p}) = 0_{\mathcal{L}^2}$. Then for $D \neq 0_{\mathbb{R}^p}$ we have:

$$g_{D_0}(D) = e^{At}(X_0 + A^{-1}(D_0 + D)) - A^{-1}(D_0 + D) - e^{At}(X_0 + A^{-1}(D_0) + A^{-1}D_0$$
$$= e^{At}A^{-1}D$$

Therefore $g_{D_0}$ is a linear function and $g$ is an affine function.

Moreover, $g$ is an injection. Indeed, if two functions are equals, then they have at most one inverse image by $g$ thanks to Cauchy-Lipschitz theorem. Therefore it defines a bijection of $\mathbb{R}^d$ in $g(\mathbb{R}^d)$. Since, $\mathcal{S}_D = g(\mathbb{R}^d)$, $\mathcal{S}_D$ is an affine space of dimension $p$ and $g$ is continuous in particular for the canonical norm induced on $\mathcal{L}^2([0,T], \mathbb{R}^p)$. Therefore $\mathcal{S}_D$ is an affine space of finite dimension and is a closed set.

**Finding a Unique Minimizer** We conclude by applying Hilbert projection lemma: our problem of minimizing $\int_0^T \|X^s(\tau) - X^D(\tau)\|$, amounts to an orthogonal projection problem. Because $\mathcal{S}_D$ is a closed convex set, we have existence and uniqueness of such projection. Therefore, it exists a unique function $X_D \in \mathcal{S}_D$ and a unique vector $D$ minimizing its distance to the function $X^s$. $\qquad \square$

## D.5 Algorithm in Linear Setting

We detail in Algorithm 2 the alternate projection algorithm in a linear setting. We denote $Y = (X_{t_0+\Delta t}^i, X_{t_0+n\Delta t}^i)$ and $X = (X_{t_0}^i, X_{t_0+(n-1).\Delta t}^i)$. For readability purposes we set $\Delta t = 1$.

---
**Algorithm 2** Alternate estimation: Linear Setting

---
**Result:** $A \in \mathcal{M}_{p,p}(\mathbb{R}), D \in \mathbb{R}^p$
$k = 0, D^0 = 0, A_0^{-1} = 0 \; A_0^0 = min_A \|\mathcal{Y} - XA\|$
  **while** $\|D^k - D^{k-1}\| > \epsilon$ *and* $\|A^k - A^{k-1}\| > \epsilon$ **do**
    $D^{k+1} = \min_D \|\mathcal{Y} - XA^k - D\|_2^2 + \lambda\|D\|_2^2$
    $A^{k+1} = \min_A \|\mathcal{Y} - XA - D^{k+1}\|_2^2 + \gamma\|Y - XA\|_2^2$
    $k \leftarrow k + 1$
**end**

---

## D.6 Proof to Proposition 3

Naturally, one could estimate jointly $D$ and $A$ using least square regression. However, the idea is to verify the convergence of such alternate algorithm in a simple case. We conduct the proof for the first dimension of $\mathcal{Y}$ to lighten notations, meaning that we are regressing the first dimension of $Y$ against the $X$.

A similar reasoning for the other dimension completes the proof.

*Proof.* We first give the analytical solution for $D$. Let $A^n$ be fixed.

**Estimation of $D$** Consider:

$$\mathcal{L}_D = \|Y - XA^n - D\|_2^2 + \lambda\|D\|_2^2 \qquad (28)$$

where $D = (d, \ldots, d) \in \mathbb{R}^Q$. For $Q$ samples, we find $d$ so that $\frac{\partial L}{\partial d} = 0$:

$$\frac{\partial L}{\partial d} = 0 \Leftrightarrow -2 * \sum_{i=1}^{Q}(y_i - X_i A^n - d) + 2\lambda d = 0$$

$$\Leftrightarrow Qd + \lambda d = \sum_{i=1}^{Q}(y_i - X_i A^n)$$

$$\Leftrightarrow d(Q + \lambda) = \sum_{i=1}^{Q}(y_i - X_i A^n)$$

$$\Leftrightarrow d = \frac{\overline{Y - XA}}{1 + \lambda/Q}$$

where $\overline{Y - XA} = \frac{1}{Q}\sum_{i=1}^{Q}(y_i - X_i A^n)$.

**Estimation of $A$**   Let $D$ be fixed and consider:

$$\mathcal{L}_A = \|Y - XA - D\|_2^2 + \gamma\|Y - XA\|_2^2 \tag{29}$$

Similarly, we aim to cancel the first derivative of $\mathcal{L}_A$ with respect to all parameters of $A = (a_1, .., a_p)$:

$$\frac{\partial \mathcal{L}_A}{\partial a_j} = 0 \Leftrightarrow -2 * \sum_{i=1}^{Q} x_{i,j}(y_i - a_0 x_{i,0} + \cdots + a_p x_{i,p} - d)$$

$$-2\gamma * \sum_{i=1}^{Q} x_{i,j}(y_i - a_0 x_{i,0} + \cdots + a_p x_{i,p}) = 0$$

$$\Leftrightarrow -2X^t(Y - XA - D) - 2\gamma X^t(Y - XA) = 0$$

$$\Leftrightarrow (1+\gamma)X^t XA - X^t(Y - D) - \gamma X^t Y = 0$$

$$\Leftrightarrow (1+\gamma)X^t XA = X^t\big(\gamma Y + (Y - D)\big)$$

$$\Leftrightarrow A = \frac{B^{-1}X^t}{1+\gamma}\big((1+\gamma)Y - D\big) \tag{30}$$

where $B = X^t X$. Equation (30) indicates that as soon a $D$ converges, $A^n$ converges. Thus, we now prove the convergence of $(D^n)$. Then, for $n > 1$ consider:

$$\big\|D^{n+1} - D^n\big\| = \frac{1}{1+\lambda/Q}\Big\|\overline{Y - XA^n} - \overline{Y - XA^{n-1}}\Big\|$$

$$= \frac{1}{1+\lambda/Q}\Big\|\overline{X(A^n - A^{n-1})}\Big\|$$

$$= \frac{1}{(1+\lambda/Q)(1+\gamma)}\Big\|\overline{XB^{-1}X^t\big([(1+\gamma)Y - D^n] - [(1+\gamma)Y - D^{n-1})]}\Big\|$$

$$= \frac{1}{(1+\lambda/Q)(1+\gamma)}\Big\|\overline{XB^{-1}X^t[D^{n-1} - D^n]}\Big\|$$

$$\leq \frac{K}{(1+\lambda/Q)(1+\gamma)}\big\|D^{n-1} - D^n\big\|$$

where $K = \|XB^{-1}X^t\|$.

Therefore, for $\lambda, \gamma$, sufficiently large, $\frac{K}{(1+\lambda/Q)(1+\gamma)} < 1$. $\|D_n - D_{n-1}\|$ converges as a positive decreasing sequence. Finally, the sequence of $(D_n)$ converge and so the sequence of $(A_n)$.

In conclusion, the proposed algorithm converges.                                                               $\square$

# E DATASETS

In this section, we provide exhaustive simulation details for the damped pendulum, Lotka-Volterra, and both geophysical datasets.

## E.1 DAMPED-PENDULUM

For the damped pendulum data, eq. (15) is integrated with $\Delta t = 0.2s$ using a Runge-Kutta 4-5 scheme from $t = 0$ up to $t = 10s$. Both the pulsation $\omega_0$ and the damping coefficient $k$ are fixed across the dataset. We generate 100/50/50 sequences respectively for train, validation and test sampling over the initial conditions so that $(\theta, \dot{\theta}) \sim \mathcal{U}\big([-\pi/2, \pi/2] \times [-0.1, 0.1]\big)$.

**Small Oscillations** To linearize the pendulum, we consider the small oscillations regime and take the initial conditions so that : $(\theta, \dot{\theta}) \sim \mathcal{U}\big([-\pi/6, \pi/6] \times [-0.1, 0.1]\big)$. In that case eq. (15) writes as:

$$\frac{d}{dt} \begin{pmatrix} \dot{\theta} \\ \theta \end{pmatrix} = \begin{pmatrix} -\lambda & \frac{g}{L} \\ 1 & 0 \end{pmatrix} \begin{pmatrix} \dot{\theta} \\ \theta \end{pmatrix} \tag{31}$$

and following notations of section 3.3, we have: $D_A = 0$ and $A = \begin{pmatrix} -\lambda & \frac{g}{L} \\ 1 & 0 \end{pmatrix}$

## E.2 LOTKA-VOLTERRA

For Lotka-Volterra data, eq. (16) is integrated with $\Delta t = 0.05$ using a Runge-Kutta 4-5 scheme from $t = 0$ up to $t = 20$. All parameters $\alpha, \beta, \gamma, \delta$ are set to 1 across the dataset. We generate 100/50/50 sequences respectively for train, validation and test sampling over the initial prey and predators populations so that $(x, y) \sim \mathcal{U}\big([0, 2]^2\big)$.

**Practical Issues and Adaptation** Assuming that $\alpha$ and $\gamma$ have positive values makes the following problem arises: the trajectories defined by $h_k$ for the prey are unbounded, whereas the trajectories defined by eq. (16) are. Minimizing $\mathrm{d}(h_k, f)$ over long term horizon will lead in an underestimation of $\alpha$ to match the bounded behaviour of true data. Therefore, we enforce $\mathrm{d}(h_k, f)$ on the prey component as soon as the number of predator is small. In practice, we set this threshold to $0.15$.

## E.3 GEOPHYSICAL DATASETS

We present in this section introductory tools for the understanding of the fluid dynamics data presented in section 4.2. We first introduce the physical modeling of ocean dynamics. Then, we outline the Adv+S dataset simulation which draws from ocean modeling. Finally, we introduce the Natl dataset and the proxy data used in the experiments.

**Introduction To Ocean Modeling** The increase in ocean observations thanks to satellites and floats enabled a great development in Earth modeling over the last decades. The ocean circulation, that is the current velocity fields dynamics, are now realistically modeled in tri-dimensional structured models such as NEMO (Madec, 2008).

Such models rely on in-depth physical knowledge of the studied system and its representation through partial differential equations. Integrated over depth, the equations associated to the transport of the Sea Surface Temperature $T$ by a time-varying horizontal velocity field $U$ can be written as:

$$\frac{\partial T}{\partial t} = -\nabla.(TU) + D^T + F^T \tag{32}$$

$$\frac{\partial U}{\partial t} = -(U.\nabla)U + f \wedge U - g'\nabla h + D^U + F^U \tag{33}$$

where $f$ is the Coriolis parameter, $h$ the depth of the surface layer obtained from sea surface height ($SSH$) observations, $g'$ the reduced gravity which takes the stratification in density of the ocean into account such that $g' \approx g.10^{-3}$. In a two-dimensional setting, $\nabla(TU)$ refers to the advection of a

scalar quantity $T$ by a velocity field $U = (u, v)$ and writes as : $\nabla(TU) = \frac{\partial T}{\partial x}u + \frac{\partial T}{\partial y}v$. The mixing terms, referred to as $D^{T/U}$ and the forcings $F^{T/U}$, are not known.

In the context of the presented work, the physical state $Z_t = (T_t, U_t)$, $f_X$ and $f_Y$ from eq. (1) can be interpreted as follows: $f_X$ represents the dynamics of the observed $T$, i.e. $f_X(T) = -\nabla.(TU) + D^T + F^T$ in eq. (32). $f_Y$ represents the dynamics of $U$ in eq. (33), i.e. $f_Y(U, h) = -(U.\nabla)U + f \wedge U - g'\nabla h + D^U + F^U$.

Whereas $T$ is observed by satellites, $U$ is not known. However, the Sea Surface Height (SSH) could be used to compute coarse estimates of $U$. Indeed, under hypothesis such as stationarity ($\frac{\partial U}{\partial t} = 0$), incompressibility ($(U.\nabla)U = 0$)), forcings can be omitted. In this case, eq. (33) can be rewritten into

$$f \wedge U = -g'\nabla h \qquad (34)$$

When projected onto $x$ and $y$ axis, eq. (34) becomes

$$-fv = -g'\frac{\partial h}{\partial x}, \qquad fu = -g'\frac{\partial h}{\partial y}, \qquad (35)$$

Note that eq. (34) and eq. (35) do not hold at fine scales as the stationarity and incompressibility assumptions only hold at large scale. In this case, the SSH $h$ can be regarded as a stream function.

Both datasets considered in the paper follow the same equations approximating the tracer equation (eq. (17)) inspired by eq. (32):

$$\frac{\partial T}{\partial t} = -\nabla.(TU) + S \qquad (36)$$

We study the equations 32 and 33 in an incremental approach. In the following parts, we describe how $T$, $U$ and $S$ are computed in both datasets Adv+S and Natl.

### E.3.1 ADV+S

We first investigate a dataset generated following simplifying assumptions (Adv+S). We don't rely on true $U$ and $S$, we instead build them so that they correspond to our hypothesis.

**Building a Velocity Field $U$**  Under stationarity and incompressibility hypothesis, $U$ can be approximated from a stream function $\mathcal{H}$. Note that, in this dataset, $\mathcal{H}$ is not equal to the SSH $h$, it is simulated following (Boffetta et al., 2001):

$$\mathcal{H}(x, y, t) = -\tanh[\frac{y - B(t) \times \cos kx}{\sqrt{1 + k^2 B(t)^2 \times sin^2 kx}}] + cy, \qquad (37)$$

As introduced precedently (see eq. (34)), eq. (33) can be simplified and we compute $U = (u, v)$ so that it follows:

$$u = -\frac{\partial \mathcal{H}}{\partial y}, \quad v = \frac{\partial \mathcal{H}}{\partial x} \qquad (38)$$

Note that $B$ varies periodically with time according to $B = B_0 + \epsilon \cos(\omega t + \phi)$. We compute 10 different velocity fields sampling random parameters $B_0, k, c, \omega, \epsilon, \phi$.

**Building a Source Term $S$**  In eq. (32), the diffusion term $D^T$ is omitted. We generate the source term $S$ so that it represents the forcing term $F^T$ in eq. (36). To illustrate heat exchanges, we draw from Frankignoul (1985). This source term is a non linear transformation of $U = (u, v)$ multiplied by the difference between the ocean temperature and a reference temperature:

$$S(U, T) = w_e \times (T - T_e) \quad \text{where} \quad w_e = \begin{cases} 0 & \text{if } \frac{\partial \mathcal{H}}{\partial t} < 10^{-4} \\ 1 & \text{otherwise.} \end{cases}$$

where $T_e$ is the sequence mean image (computed without source).

**Dataset Generation**  Using computed $U$ and $S$, we integrate eq. (36) with $\Delta t = 8640s$ over 30 days, using a Semi-Lagrangian scheme (see explanations below). We generate 800/100/200 sequences respectively for train, validation and test sampling over the initial conditions, which are images of size $64 \times 64$ sampled from Natl dataset. Finally, for integration, we impose East-West periodic conditions, implying that what comes out the left part re-enters at the right, and reciprocally. We also impose velocity to be null on both top and bottom parts of the image.

**Semi-Lagrangian Integration**  Unlike Eulerian scheme, relying on time discretization of the derivative, the semi Lagrangian scheme relies on the constancy of the solution of a PDE along *a characteristic curve*. Consider a solution to the advection equation, i.e. eq. (36) with $S = 0$. The method of characteristics consists in exhibiting curves $(x(s), t(s))$ along which the derivative of the solution $T$ is simple, i.e $\frac{\partial T}{\partial s}(x(s), t(s)) = 0$. For a 1D constant advection scheme, computations lead to:

$$\frac{dt}{ds} = 1 \implies s = t$$
$$\frac{dx}{ds} = U \implies x = x_0 + Ut$$

giving therefore, $T(x, t) = T_0(x - Ut)$, linking the value of the solution at all time to its initial condition. Therefore from a single observation at $t_0$, it suffices to estimate the original departure points $x_0 - Ut$ to infer the prediction at $t$.

However, when $U$ is not constant in time, the method remains doable, not along characteristic *lines* defined by : $(x_0 + Ut)$, but along characteristic *curves* which are given by:

$$\frac{dt}{ds} = 1 \implies s = t$$
$$\frac{dx}{ds} = U(x, t) \tag{39}$$

A great deal in the semi-Lagrangian literature involves solving correctly eq. (39). We use the conventional mid-point integration rule and the semi-Lagrangian is implemented using Pytorch function `gridsample`, following in (Jaderberg et al., 2015). Further developments can be found for example in (Diamantakis, 2014).

### E.3.2  NATL

This second dataset depicts the actual ocean circulation, i.e. we consider both eq. (32) and eq. (33). In this case, no assumptions are made on $U$ and $S$ represents both diffusion terms $D^T$ and forcing terms $F^T$. We access daily data over a year of ocean surface temperature of the North Atlantic observations model resulting from (Ajayi et al., 2019) [1]. The dataset covers a 2300km $\times$ 2560km zone at 1.5km resolution, in the North Atlantic Ocean.

In this real-life dataset, sea surface height (SSH) partial derivative with respect to $x$ and $y$ serves as proxies to the (unobserved) velocity fields $U$. Indeed, recall that simplifying hypotheses led us to eq. (35).

We divide the Natl zone into 270 patches of size $64 \times 64$. For each region, we extract sea surface temperatures, velocity fields, source terms and height variables. We sample 200/20/50 sequences of 1 year, for respectively train, validation and test. In this case, $\Delta t = 86400s$ (1 day).

## F  TRAINING DETAILS

All experiments were conducted on NVIDIA TITAN X GPU using Pytorch (Paszke et al., 2019).

**Hyper-Parameters Interpretation**  From eq. (4), two independent terms appear justifying an alternate projections approach.

First, we highlight that strictly minimizing $\mathrm{d}(h_k, f)$ biases our estimation of $h_k$. However, it may yield a good estimation of $h_k$ provided that $f_k$ contributes significantly to the prediction of $f$. Hence, we interpret this loss as an *initialization* loss. Thus, in most applications, we progressively decrease its magnitude along training as detailed in appendices F.1 to F.3.

On the other hand, $\mathrm{d}(h_u, 0)$ aims at constraining the free form function $h_u$ to make its action as small as possible. We interpret this loss as a *stability* penalty.

---

[1]Details available at : https://meom-group.github.io/swot-natl60/access-data.html

Finally, aiming to recover exact trajectories of observations, we proceed as suggested in (Yin et al., 2021) progressively increasing the hyper-parameters associated to $\mathrm{d}(h, f)$.

The practical implementation is summarized in the following algorithm:

---

**Algorithm 3** Alternate estimation: Practical Setting

---

Initialization: $\theta_u^0 = 0$, $\theta_k^0 = \min_{h_k \in \mathcal{H}_k} \mathrm{d}(h_k, f)$, $\lambda_h, \lambda_{h_k}, \lambda_{h_u}$
**for** $epoch = 1 : N_{epochs}$ **do**
    **for** $batch = 1 : B_k$ **do**
$$\theta_k^{n+1} = \theta_k^n - \tau_1 \nabla_{\theta_k}[\lambda_h \mathrm{d}(h, f) + \lambda_{h_k} \ell(h_k)]$$
    **end**
    **for** $batch = B_k : B_u$ **do**
$$\theta_u^{n+1} = \theta_u^n - \tau_1 \nabla_{\theta_u}[\lambda_h \mathrm{d}(h, f) + \lambda_{h_u} \mathrm{d}(h_u, 0)]$$
    **end**
    $\lambda_h = \tau_2 \lambda_h$; $\lambda_{h_k} = \frac{1}{\tau_2} \lambda_{h_k}$; $\lambda_{h_u} = \frac{1}{\tau_2} \lambda_{h_u}$
**end**

---

## F.1   Damped Pendulum

**Architecture Details**   The physical parameters to be learned is a scalar of dimension 1, and $h_u$ is a 1-hidden layer MLP with 200-hidden neurons with leaky-relu activation.

**Optimization**   For this dataset we use RMSProp optimizer with learning rate 0.0004 for 100 epochs with batch size 128. We supervise the trajectories up to $t = \Delta t \times 50$, i.e we enforce $\mathrm{d}_\phi$ over $(t_0 + \Delta t, .., t_0 + 50\Delta t)$. Overall the number of optimization subsequences for training is 17000. We alternate projection on $\mathcal{S}_k$ and $\mathcal{S}_u$ by descending the gradient 10-batches on $h_k$ then 10-batches on $h_u$.

**Hyperparameters**   We initialize $\lambda_{h_k} = 0.1$ and decrease it geometrically down to $\lambda_{h_k} = 0.001$. We initialize $\lambda_h = 0.1$ and increase it geometrically up to $\lambda_h = 100$. $\lambda_{h_u}$ is fixed through training at $0.1$.

The hyper-parameters were chosen by randomly exploring the hyper-parameters space by sampling them so that $\lambda \sim \mathcal{U}(1, 0.1, \ldots, 10^{-4})$. We select the ones with the lowest prediction errors, i.e with lowest $\mathrm{d}_\phi(h, f)$.

For the ablation study of Table 1, we set to 0 the hyper-parameters associated to the non-considered loss.

The training time for this dataset is 1 hour.

## F.2   Lotka-Volterra

**Architecture Details**   The physical parameters to be learned is a vector of dimension 2 accounting for $(\alpha, \beta)$ in eq. (16), and $h_u$ is a 1-hidden layer MLP with 200-hidden neurons with leaky-relu activation.

**Optimization**   We use Adam optimizer with learning rate 0.0005 for 200 epochs with batch size 128. Overall the number of sequences for training is 15000. We supervise the trajectories up to $t = \Delta t \times 25$, i.e we enforce $\mathrm{d}_\phi$ over $(t_0 + \Delta t, .., t_0 + 25\Delta t)$. We alternate projection on $\mathcal{S}_k$ and $\mathcal{S}_u$ by descending the gradient 10-batches on $h_k$ then 10-batches on $h_u$.

**Hyperparameters**   We initialize $\lambda_{h_k} = 0.1$ and decrease it geometrically down to $\lambda_{h_k} = 0.001$. We initialize $\lambda_h = 0.001$ and increase it geometrically up to $\lambda_h = 1$. $\lambda_{h_u}$ is fixed through training at $0.001$.

The hyper-parameters were chosen by randomly exploring the hyper-parameters space by sampling them so that $\lambda \sim \mathcal{U}(1, 0.1, \ldots, 10^{-4})$. We select the ones with the lowest prediction errors (i.e lowest $d(h, f)$).

For the ablation study of Table 1, we set to $0$ the hyper-parameters associated to the non-considered loss.

The training time for this dataset is 2 hours.

### F.3  ADV+S

**Architectures Details**  The physical parameters to be estimated are the velocity fields $U$, of dimension $(2, 64, 64)$. As $U$ varies over time, we follow data assimilation principles to map a sequence of 4 consecutive measurements of the tracer field $T$ to the associated velocity field (Gaultier et al., 2013). To do so, we parameterize a recognition network $G_\psi$ by U-net with at most 512 latent channels also following the implementation of (Isola et al., 2017), taking as input a sequence of 4 time steps of $T$: $(T_{t_0}, .., T_{t_0+3\Delta t})$. The residual dynamics $h_u$ is learned by a convolutional ResNet, with 1 residual block taking as entry the same sequence of $T$. We implement $h_k$ via a semi-lagrangian scheme, taking as input $T_t$ and the estimated $U_t$ to predict $T_{t+1}$.

**Optimization**  We use Adam optimizer with learning rate $0.0001$ for 30 epochs with batch size 32. We supervise the trajectories up to $t = \Delta t \times 6$, i.e we enforce $d_\phi$ on $(T_{t_0+\Delta t}, ..., T_{t_0+6\Delta t})$. Overall the number of sequences for training is $36800$. We alternate projection on $\mathcal{S}_k$ and $\mathcal{S}_u$ by descending the gradient 4-batches on $h_k$ then 6-batches on $h_u$.

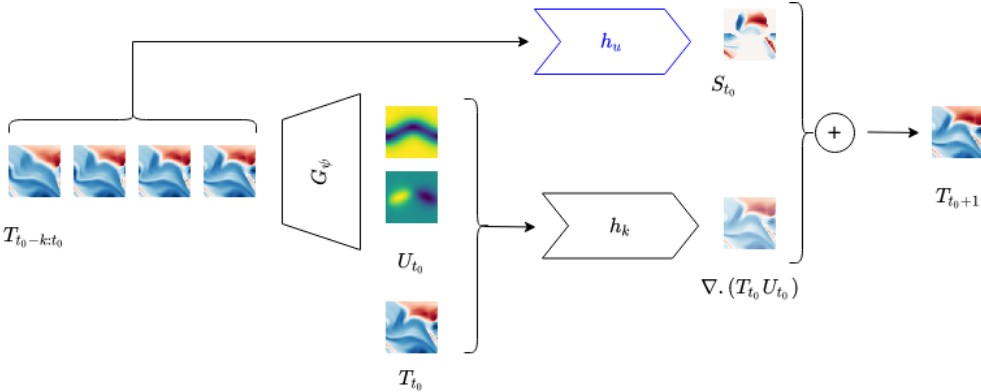

Figure 3: *Best viewed in color.* Schematic view of our model in the context of section 5.2, applied on the Adv+S dataset.

**Hyperparameters, setting of eq. (4)**  We initialize $\lambda_{h_k} = 0.1$ and decrease it geometrically down to $\lambda_{h_k} = 0.00001$. We initialize $\lambda_h = 0.01$ and increase it geometrically every epoch up to $\lambda_{h,f} = 1000$. $\lambda_{h_u}$ is fixed through training at $0.1$. We select the hyperparameters with the lowest prediction errors (i.e lowest $d(h, f)$). For the ablation study of Table 1, we set to $0$ the hyper-parameters associated to the non-considered loss.

The training time for this dataset is 8 hours.

### F.4  NATL

**Architecture Details**  The architectures in this setting are identical to the ones described in appendix F.3.

**Optimization**  We use Adam optimizer with learning rate $0.00001$ for 50 epochs with batch size 32. Overall the number of sequences for training is $67000$. We enforce $d_\phi$ over 6 time-steps, i.e we supervise the predictions on timesteps: $(t_0 + \Delta t, .., t_0 + 6\Delta t)$. We use dropout in both $G_\psi$ and $h_u$.

**Hyperparameters, setting of eq. (4)** For this setting, $\lambda_h$ geometrically increases from 0.01 up to 100. We initialize $\lambda_{h_k} = 0.1$ and decrease it geometrically down to $\lambda_{h_k} = 0.00001$. $\lambda_{h_u}$ is fixed through training at 0.1. We alternate projection on $\mathcal{S}_k$ and $\mathcal{S}_u$ by descending the gradient 10-batches on both $h_k$ and $h_u$.

The selected model is the one with lowest prediction errors on validation set (i.e lowest $\mathrm{d}(h, f)$), sampling uniformly the hyperparameters: $\lambda \sim \mathcal{U}(1, 0.1, \ldots, 10^{-4})$.

**Hyperparameters, setting of eq. (5)** Because the dynamics of Natl is highly non linear and chaotic, we follow (Jia et al., 2019) and first warm-up the parameters recognition network $G_\psi$ on the velocity fields proxies for 10 epochs. For this setting, $\lambda_h$ geometrically increase from 0.01 up to 1. $\lambda_{h_k}$ is set equal to $\lambda_h$. $\lambda_{h_u}$ is fixed through training at 0.01.

After warm-up, we alternate projection on $\mathcal{S}_k$ and $\mathcal{S}_u$ by descending the gradient 100-batches on $h_k$ and 300 on $h_u$. In this setting of eq. (5), the selected model is the one with lowest $\mathrm{d}(h, f) + \mathrm{d}(h_k, f_k^{pr})$ error, sampling uniformly the hyperparameters: $\lambda \sim \mathcal{U}(1, 0.1, \ldots, 10^{-4})$.

The training time for this dataset is 12 hours.

**Baselines** We train NODE (Chen et al., 2018) and Aphynity (Yin et al., 2021) on both the Adv+S and Natl dataset. For the training of Aphinity, we set the learning rate at 0.0001 and train on 30 epochs. We initialize $\lambda_h = 0.01$ and increase it geometrically every epoch up to $\lambda_h = 100$. $\lambda_{h_u}$ is fixed through training at 0.1. For the training of NODE, we set the learning rate at 0.00004 and train on 50 epochs. To perform prediction, we first encode the 4-consecutive measurements of $T$ (as a $3 \times 64 \times 64$ state) then learn to integrate this state in time thanks to a network $h$. $h$ is a 3-layer convolutional networks, with 64 hidden channels. It is integrated using RK4 scheme available from `https://github.com/rtqichen/torchdiffeq`.

## G ADDITIONAL RESULTS AND SAMPLES

### G.1 RESULTS FOR PENDULUM AND LOTKA-VOLTERRA DATASETS

We provide respectively in figs. 4 and 5 phase diagrams for the damped pendulum and Lotka-Volterra experiments. Both graphs in the phase space indicate that the trajectories and their nature are well handled by the learned decomposition, providing a periodic phase space for Lotka-Volterra (fig. 5), and a converging spiral for the damped pendulum (fig. 4).

### G.2 RESULTS FOR ADV+S AND NATL

In this section, we provide additional results on both Adv+S and Natl datasets. A thorough ablation study (table 4) gives results with constant hyperparameters $\lambda_h$ and $\lambda_{h_k}$ (row Vanilla Optim), which validates our hyper-parameters interpretation. Indeed, the results are better when respectively increasing and decreasing $\lambda_h$ and $\lambda_{h_k}$. Besides, the row Ours eq. (5) refers to a training performed as introduced in appendix C.4 with $f^{pr} = h^\star$ trained on eq. (4). Figure 7 shows predictions up to 4 days on the Adv+S data. Finally, figs. 9 and 11 provide results on Natl dataset associated to training relying on both eq. (4) and eq. (5) and with NODE (Chen et al., 2018).

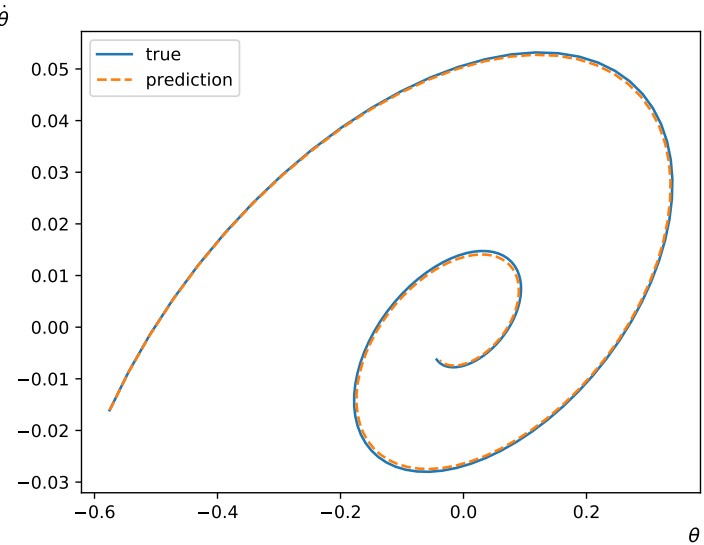

Figure 4: Damped Pendulum Phase Diagram. The true phase diagram (blue) and learned (orange dashed) are close, indicating consistency in the prediction

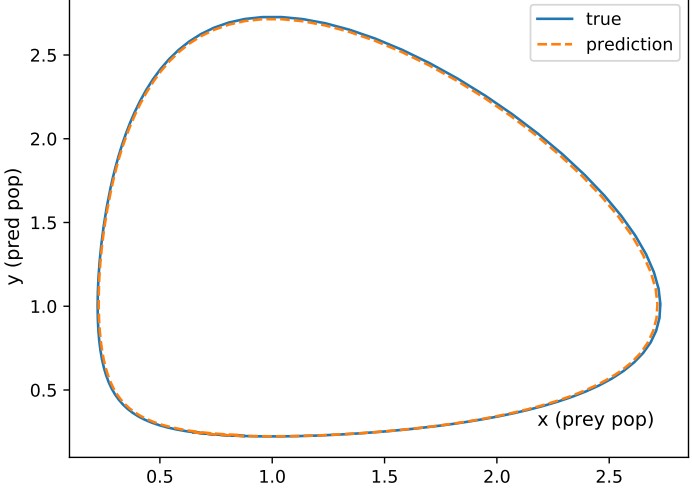

Figure 5: Lotka-Volterra Phase Diagram. The true phase diagram (blue) and learned (orange dashed) are close, indicating consistency in the prediction

Table 4: Ablation Study on Adv+S. We report the MSE ($\times$ 100) on the predicted observations $T$, the estimated velocity fields $U$ and the residual source term $S$ over 6 and 20 time steps from an initial datum $t_0$. Unlike alternate training, i.e. Algorithm 1, "Joint" rows refer to the simultaneous optimization of $h_k$ and $h_u$.

| Training | Models | $t_0 + 6$ | | | $t_0 + 20$ | | |
|---|---|---|---|---|---|---|---|
| | | $T$ | $U$ | $S$ | $T$ | $U$ | $S$ |
| | Ours ($U$ known) | 0.52 | n/a | 0.19 | 2.0 | n/a | 0.32 |
| Alternate | Ours eq. (4) | **0.74** | **1.99** | 0.17 | 8.49 | **2.26** | 0.31 |
| | only $\mathrm{d}(h, f)$ | 1.02 | 4.08 | 0.19 | 10.59 | 4.19 | 0.32 |
| | $\mathrm{d}(h, f) + \mathrm{d}(h_k, f)$ | 1.02 | 3.66 | 0.19 | 11.42 | 3.84 | 0.34 |
| | $\mathrm{d}(h, f) + \mathrm{d}(h, h_k)$ | 0.77 | 2.38 | 0.19 | 9.5 | 2.45 | 0.34 |
| | Ours eq. (5) | 0.75 | 2.77 | **0.17** | **8.36** | 2.84 | **0.29** |
| | Vanilla optim. | 1.51 | 3.77 | 0.3 | 13.33 | 4.1 | 5.15 |
| Joint | Ours eq. (4) | 1.44 | 3.3 | 0.3 | 12.82 | 3.5 | 0.5 |
| | only $\mathrm{d}(h, f)$ | 1.38 | 6.96 | 0.39 | 11.9 | 7.09 | 0.54 |

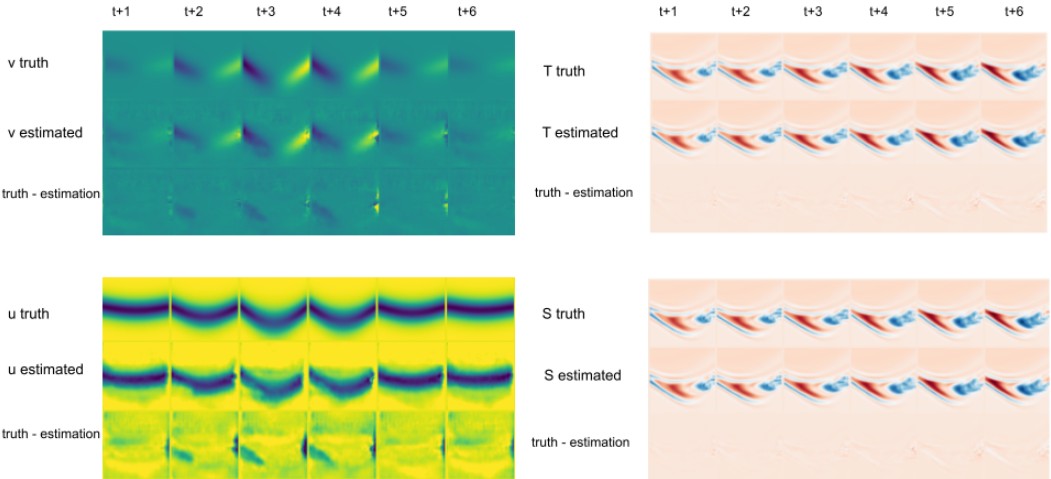

Figure 6: *Best viewed in color.* Estimations, targets and differences between estimations and targets on $T$, $U = (u, v)$ and $S$ for Adv+S. Each column refers to a time step, ranging from 1 to 6 half-days. On the left, true and estimated $U = (u, v)$ over 6 time steps, and differences between targets and estimations. On the right, prediction of $T$ and $S$ over 6 time steps, and differences between targets and estimations.

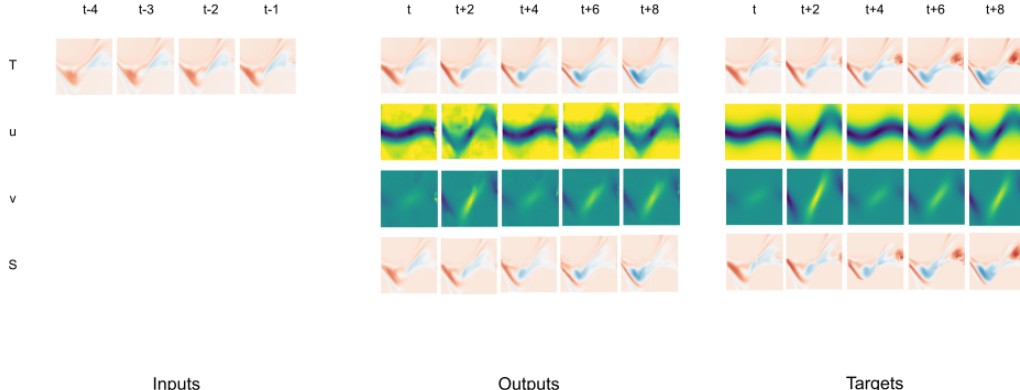

Figure 7: *Best viewed in color.* Estimations and targets on $T$, $U = (u, v)$ and $S$ for Adv+S. Each column refers to a time step, ranging from 1 to 8 half-days. On the left, sequence of $T$ inputs (4 time steps). In the middle, prediction of $T$, $U = (u, v)$ and $S$ over 8 time steps. On the right, true $T$, $U$ and $S$ over 8 time steps.

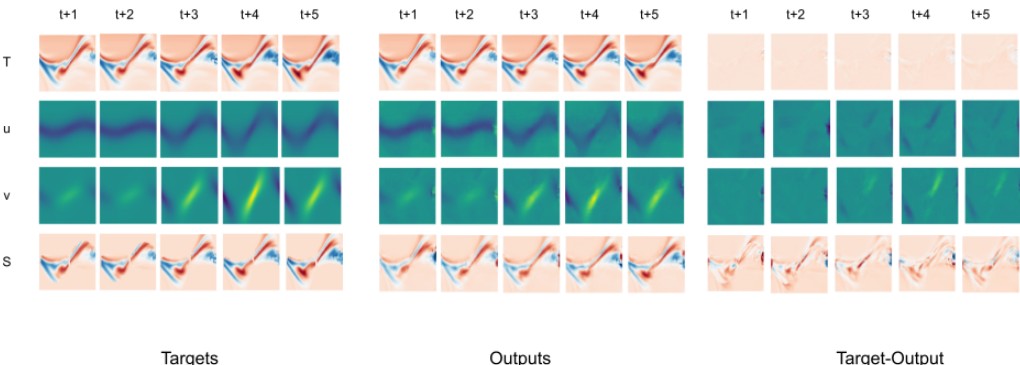

Figure 8: *Best viewed in color.* Estimations, targets and differences between estimations and targets on $T$, $U = (u, v)$ and $S$ for Adv+S. Each column refers to a time step, ranging from 1 to 5 half-days. On the left, true $T$, $U$ and $S$ over 5 time steps.. In the middle, prediction of $T$, $U = (u, v)$ and $S$ over 8 time steps. On the right, differences between targets and estimations.

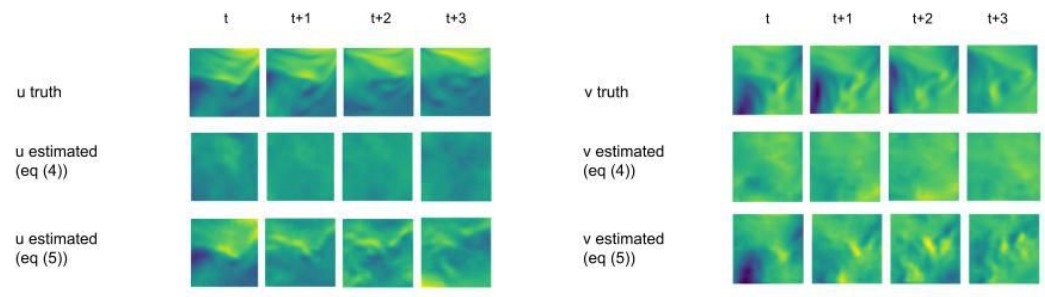

Figure 9: *Best viewed in color.* Sequence of estimations on $U = (u, v)$ for the Natl data. The second and third row respectively refer to training according to eq. (4) and eq. (5). The loss term $\mathrm{d}(h_k, f_k^{pr})$ in eq. (5) enables our model to learn more accurate velocity fields than when only trained following eq. (4).

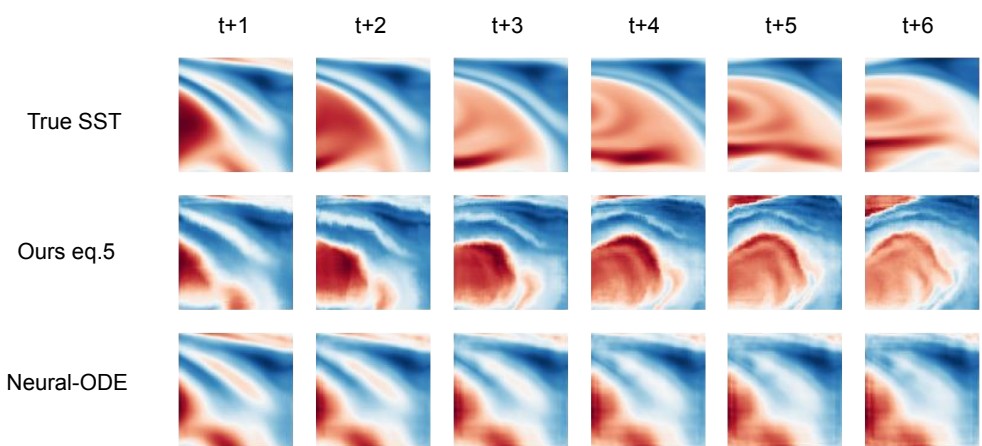

Figure 10: *Best viewed in color.* Sequence of prediction on $T$ for the Natl data. Contrary to our model (row eq. (5)), NODE (row Neural-ODE) struggles to predict any motion in $T$.

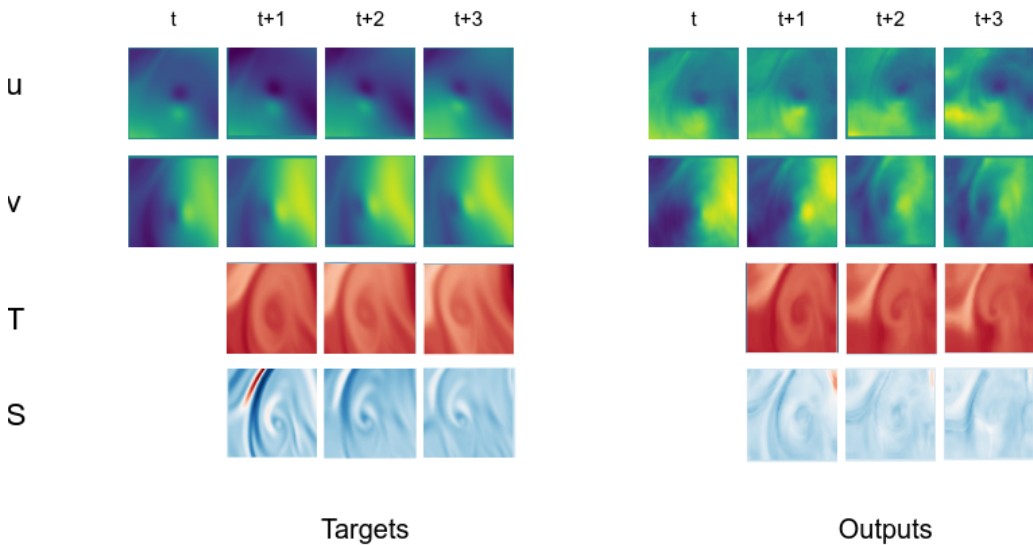

Figure 11: *Best viewed in color.* Sequence of prediction on $T, u, v, S$ for the Natl data across 3 days trained using proxy data according to eq. (5)

