# OpenReview forum: "Constrained Physical-Statistics Models for Dynamical System Identification and Prediction"
_ICLR.cc/2022/Conference — ICLR 2022 Poster_

### Official Review · Reviewer_GA76 · 2021-10-26

**Correctness:** 4
**Technical Novelty And Significance:** 2
**Empirical Novelty And Significance:** 2
**Recommendation:** 3
**Confidence:** 3

**Main Review:**

Strength: The main strength of this paper is the experiments. In the experiment, the results is promising, with better performance in both synthetic and real-world experiments. Ablation studied are also performed to show the model performance with different objective functions.

Weak: I think the weakness of this paper is the technical novelty of the algorithm and theorem. The idea of combining physical models and machine learning models exist for decades. The theoretical derivations in this paper is nothing but applying triangle inequalities here and there. The alternate minimization algorithm is not novel at all, and the convergence of such an algorithm for a linear dynamic system is also well-known. Therefore, the technical contribution of this paper seems trivial.

**Summary Of The Paper:**

This paper investigate a novel method to recover well-posedness and interpretability for the problem of learning a physical and machine-learning hybrid model, by controlling the ML component and the MB hypothesis. The authors also study the case where auxiliary data are introduced. Experiments show the performance of the proposed method in both synthetic data and real-world ocean dynamics.

**Summary Of The Review:**

The experiment results of this paper is very strong and comprehensive. However, the novelty of this paper is not very good. The motivation, theoretical analysis and algorithms are not very novel.

---

> ### Author Response · Authors · 2021-11-16
> **Answer to R.GA76**
>
> We respectfully but firmly disagree with your claims regarding our contributions and would be glad to answer any precise question regarding any technical aspects of our work and the originality of our approach. Below are some facts that might help to remove this misconception.
>
> ## On the novelty of combining physical and machine learning models
>
> The positioning of the domain is clearly described in the introduction and the related work sections of the paper. Let us rephrase some of the arguments presented here. Although there have been some contributions before, the topic of physics informed machine learning has witnessed an unprecedented research effort over the last 3 years with an increasing number of publications in fields as diverse as machine learning, physics, numerical analysis, CFD. This is a very timely topic with considerable stakes both at the academics and industrial levels. You might be interested in the recent Nature paper [1] (2019), or the recent survey on physics based ML [3] (2021), or the recent monograph on Physics-based deep learning [2] (2021). Note that there exists also regular workshops at top ML conferences that testify the novelty and the dynamism of this emerging field.
>
>
> ## Regarding the technical contributions
>
> We answer to the two main criticisms:
>
> “The theoretical derivations in this paper is nothing but applying triangle inequalities here and there”
>
> We feel extremely disappointed by this comment. First, we emphasize that the relevance of a method and the tools it mobilizes are two separate things. In that perspective, we prove some strong and original results concerning the decomposition of physical-statistical models for dynamical systems. Since part of our contribution relies on bounds, we indeed use inequalities, but a simple look at the 6 pages of proofs in the supplementary clearly shows, in our opinion, that the comment is not relevant.
>
> “The alternate minimization algorithm is not novel at all, and the convergence of such an algorithm for a linear dynamic system is also well-known”
>
> Indeed, any machine learning algorithm relies on an optimization technique. Here we make use of alternate minimization and we explain why this is relevant in our case. Of course, it is never claimed in this paper that alternate minimization is a contribution per se. After all, gradient descent is a very longstanding mathematical tool (I.Newton, T. Simpson, A.Cauchy ...). As for the convergence, it is demonstrated for solving eq. (13) and (14) as a necessary sanity check.
>
>
> [1] Deep learning and process understanding for data-driven Earth system science. _Reichstein, M., Camps-Valls, G., Stevens, B., Jung, M., Denzler, J., Carvalhais, N. and Prabhat 2019._ Nature. 566, (2019), 195–204.
>
> [2]  Physics-based Deep Learning, _Thuerey, N., Holl, P., Mueller, M., Schnell, P., Trost, F. and Um, K_ http://arxiv.org/abs/2109.05237 (2021)
>
> [3] Integrating physics-based modeling with machine learning: A survey.  _Willard, J.D., Jia, X., Xu, S., Steinbach, M. and Kumar, V._ arXiv (2021), 1–34.

---

### Official Review · Reviewer_1AX7 · 2021-11-01

**Correctness:** 4
**Technical Novelty And Significance:** 3
**Empirical Novelty And Significance:** 3
**Recommendation:** 8
**Confidence:** 4

**Main Review:**

This paper is very well written and the problem being addressed is very relevant to the machine learning community, especially to its emerging applications in science and engineering problems. The proposed method is sound and quite easy to implement, and the performance is also good as presented. I think it will be of interest to the broad readership on the intersection of machine learning and scientific applications.

Below, I have some questions that the authors may wish to clarify.

1. The overall approach: The key method introduced to ensure well-posedness is page 4 Eq (7), via constraining each part of the model by $\mu_u,\mu_k$. However, to me it is often more convenient in practice to consider the “dual” formulation of optimization with penalization, i.e. instead of (7) one can consider
   $$
   \min_{h_u,h_k} d(f, h_u+h_k) + \lambda_k \ell(h_k) + \lambda_u d(h_u,0)
   $$
   This is certainly very easy to implement, and has the same number of hyper-parameters as the proposed case, replacing the $\mu$‘s by $\lambda$’s. Can the authors comment on why the current constrained approach is chosen over this approach, and can some quantitative comparisons be made?

2. Algorithm 1: while successive projections to convex sets can find the intersection eventually, the convergence rate may be slow. Partial projections can sometimes speed up the process. Do you observe speedups if equation (8) is not solved to completion in each step?

3. Prop 3: $D$ is proved to converge (and thus $A$) via a contraction argument. Can anything be said about the convergence point? Can it be a local minimum but not global?

4. Table 1: As I understand, the performance of the proposed method will depend on the choice of the hyper-parameters $\mu_k,\mu_u$. Can the authors comment on the sensitivity of tuning parameters with respect to the good results shown in this table? How is fair comparison ensured, e.g. how are the baseline method(s) (e.g. Yin et al) tuned to each problem?

5. Section 4.2: This example is interesting as it is more realistic for scientific applications. I believe this part has some clarity issues in the main text

   1. I believe that Eq (32-33) in the appendix should be included in the main text to allow for easier understanding of what is $f_k$ and $f_u$.
   2. Looking at Alg 3 in the appendix, the alternate minimization is in fact done by gradient descent with soft constraints instead of the alternative projections introduced in Alg 1 in the main paper. Can the authors explain why this modification is made, and whether the prior experiments on toy examples followed Alg 1 or Alg 3?

**Summary Of The Paper:**

This paper studies the identification and prediction of dynamical systems from observed data by combining statistical approaches with physical priors. The main idea is to decompose the approximation of unknown dynamics into two parts: a parametric physics-based model $h_k$ (used to incorporate prior knowledge, or to identify parameters), and a non-parametric statistical model $h_u$ (used to fit the unknown, or unobserved degrees of freedom).

Since the statistical part has a priori no restrictions, the overall problem is ill-posed in the sense that the statistical part can fit all of the dynamics. To ensure well-posedness, the authors propose to convert the joint learning problem to learning $h_k$ and $h_u$ subject to constraints: $h_k$ is given an allowed error tolerance and $h_u$ is given an allowed deviation from 0. The authors show that this modification implies well-posedness, and subsequently design an alternative minimization algorithms to solve for $h_k$ and $h_u$ iteratively. It is proved that this algorithm converges in the affine setting (via sequential projection to find intersection of convex sets) and it is demonstrated experimentally in low and high dimensional systems that this method works well in practice outside of the convex setting.

**Summary Of The Review:**

This is a good paper tackling a relevant problem. The algorithm introduced is easy to implement and has promising performance. Some theoretical justification on convergence is also presented. I believe that this is a meaningful contribution and should be accepted. There are some clarity issues that the authors can address in a revision.

---

> ### Author Response · Authors · 2021-11-16
> **Answer to R.1AX7**
>
>
> We thank you very much for your positive and valuable review.
>
>
> 1. The introduction of $\mathcal{S}_k$ and $\mathcal{S}_u$ in the constrained (primal) formulation are required in order to be able to derive the theoretical properties of the proposed framework (proposition 1). It also enables us to define the projection onto the sets $\mathcal{S}_k$ and $\mathcal{S}_u$. However, as duly noted, for a practical implementation with NN, the Lagrangian (dual) formation is better suited. We use this dual formulation in the “practical optimization 3.2.2” section. The corresponding “dual” algorithm is described in supplementary section F. algorithm 3. We have modified the text to make clear that Algorithm 1 is only a generic method and that the practical algorithm is indeed a dual formulation as described in section 3.2.2.  We do hope that this clarifies the point you highlighted.
>
> 2. You are perfectly right, partial descent is effective for our setting, and this is what is done here. We rely on the SGD optimization of eq.8, and the number of steps required to optimize each component is a hyper parameter. Computing the actual projection requires many gradient steps, but in practice, we found out that using simple partial projection schemes and descending the gradient $10$-steps on $h_k$ then $10$-steps on $h_u$ provides good results for PDL and LV data.
>
> 3. Unfortunately, it is difficult to characterize in this setting the convergence point since $f$ is not necessarily a linear function (see discussion with Reviewer YsB3, on proposition 1 - 1st item in our answer). If one assumes that $f$ is linear and that the residuals follow a normal distribution, we could make use of linear regression theory and state that the estimation of $A$ will follow a distribution centered around the true $A$. For nonlinear settings, the convergence point can only be judged empirically.
>
> 4. Hyperparameters were classically chosen to minimize the prediction loss on a validation set via random sampling as described in the appendix F. As explained before, we worked with the implementation of the dual formulation, with hyperparameters the $\lambda$. There is no manual exploration of this hyperparameter space. The experimental setting is similar for the baselines. The method used for training the model of [1] follows the algorithm presented in the original paper (with the tuning of $\tau$, and $N_{iter}$ on a validation set, but keeping approximately the same number of gradient steps for the two methods, for a fair comparison).
>
>
>
> 5.
>     - Eq.(17) presents a simplification of eq.32-33, with alleviated notations. Also, eq 32-33 only refers to Natl experiments and requires the introduction of several variables with complex definitions that do not fit easily in the main text and make the reading more difficult. Thus, we chose to provide  them in the supplementary for the sake of readability.
>     - Sorry for the misunderstanding, we hope that this is now clarified. All the experiments are conducted using algorithm 3, i.e the gradient descent based version of algorithm 1.
>
>
> [1] Augmenting Physical Models with Deep Networks for Complex Dynamics Forecasting _Vincent Le Guen, Yuan Yin, Jérémie Dona, Ibrahim Ayed, Emmanuel de Bézenac, Nicolas Thome, Patrick Gallinari_, ICLR 2021

---

> > ### Comment · Reviewer_1AX7 · 2021-12-02
> > **Reply**
> >
> > I'd like to thank the authors for addressing my questions. I will keep the current score.

---

### Official Review · Reviewer_TpQa · 2021-11-02

**Correctness:** 4
**Technical Novelty And Significance:** 3
**Empirical Novelty And Significance:** 2
**Recommendation:** 6
**Confidence:** 4

**Main Review:**

Authors propose a unifying framework for hybrid models, generalizing recent methods. They derive theoretical guarantees very compelling theoretical guarantees for the framework, as well as a convergence proof for a simple affine model. However, I believe that the paper is quite hard to read, and propose several models without giving intuition for which model to use. The models also seem to need complex hyper-parameter choice, which again is not very intuitive and not discussed extensively in the supplementary material.

Here are two points that I think need to be discussed more extensively in the paper:
 - The paper proposes two models (with and without auxiliary data). When should we use one or the other? Should we just use it whenever auxiliary data is available? Moreover, equation 5 is the equivalent of equation 4 for auxiliary data. However, the model uses alternate optimization to converge. What is the equivalent procedure for auxiliary data?
 - The "practical optimization" is not straightforward, and although it is well-detailed, the hyper-parameters chosen for the experiments are pretty complex, which is to me an important drawback of the framework. To give intuition into the model, how do the lambda values relate to the two scalars mu in equation 6?

Although the section titled "THEORETICAL ANALYSIS FOR A LINEAR APPROXIMATION" is compelling, I think it could be included in the appendix to allow for more space to discuss the previous points.

Other points that would make the paper easier to read :
 - Using "Ours eq 4" in the results is not intuitive at all, since the model optimizes a different objective. This makes it a bit confusing.
 - Figures are not rendered properly, many of them appear blurry and could be made bigger (there are lots of white space in the figure that could be removed). This greatly affects the quality of the paper.
 - I would like to see the differences between true and estimated processes in the figures. This would make the results easier to grasp.

The method is compared to Aphynity and Neural ODE but there are no comparisons to Neural ODE on the simulated datasets (I think it would be beneficial to show the predictive logMSE for Neural ODE in Table 1).



**Summary Of The Paper:**

Authors propose a general framework for learning and ensuring identifiability with hybrid models. They derive strong theoretical guarantees as well as a proof of convergence in a simple affine setting. They validate their framework in several simulated experiments and the Natl dataset, and draw comparisons with state-of-the-art hybrid model Aphynity as well as Neural ODEs. Experiments confirm that the proposed framework increase interpretability and maintain high prediction performances

**Summary Of The Review:**

Authors propose a unifying framework for hybrid models, generalizing recent methods. They derive theoretical guarantees very compelling theoretical guarantees for the framework, as well as a convergence proof for a simple affine model. However, I believe that the paper is quite hard to read, and propose several models without giving intuition for which model to use. The models also seem to need complex hyper-parameter choice, which again is not very intuitive and not discussed extensively in the supplementary material.

Thus, I do not recommend acceptance of this paper, but I am willing to increase my score if authors answer my concerns, as the theoretical guarantees derived here are very strong and surely beneficial for a large class of hybrid models.

---

> ### Author Response · Authors · 2021-11-16
> **Answer to R.TpQa (1/2)**
>
> We thank you for your insightful review.
>
> ## About auxiliary data
> We are sorry if we were not clear enough on this topic. Most work in the current literature only addresses learning of simple dynamics simulated from generic PDEs and does not consider real world data. There is a huge gap between the two settings.
> In the same spirit as what is done in the literature, we first address ideal use cases of increasing difficulty (pendulum, Lotka-Volterra, and a generic advection equation).  In the paper, eq. 4 corresponds to the idealized setting and has been developed as a first step towards more real situations.
> Then we tackle a far more complex dynamics, Natl, corresponding to real ocean dynamics.
> Our observation, stemming from our experience with different datasets, is that for complex problems, close to real use cases, auxiliary data or information are required in order to get physically sound solutions. Without such additional information, it is usually impossible to learn the complex dynamics using a pure data driven approach. In order to incorporate this information, we developed the bound for eq.5. Notably, the way to derive the constraints of eq.5 slightly differs from the derivation of eq.4. Incorporating this information,  by exploiting proxy data,  provides significant improvements, as  illustrated on table 2 for the Natl data: purely data driven models and a model incorporating auxiliary information are compared. A reason for such gain in performances is that the model estimating the $\theta_k$ now improves over the existing knowledge instead of training from scratch as in the setting developed in eq.4.  Moreover, this setting provides more data to train on, leading to a more robust estimation.
>
> The optimization procedure defined by Algorithm (1) encompasses both learning settings. In the practical optimization section 3.2.2 $\ell(h_k) =  \mathrm{d}(h_k,f)$ in the case of eq.4 and  $\ell(h_k) =  \mathrm{d}(h_k,f_k ^{pr})$ in the case of eq.5.
>
> We clarified this in section 3.1.1 and 3.1.2 of the main text.
>
>
> ## Practical optimization
> In the main text, we do not provide details on how to set up the parameters, but we give more information in Supplementary Section F. Overall, this is a classical hyperparameter search like in most DL implementations. We summarize here the main findings. (i)  $\mathrm{d}(h_k,f)$ can be interpreted as an initialization loss, that aims at yielding a first good guess for $h_k$, (ii) $\mathrm{d}(h_u,0)$ is interpreted as a stability penalty that aims at containing the contribution of the statistical component. (i) means that the coefficient $\lambda_{h_k}$ before $\mathrm{d}(h_k,f)$ should be decreased during training. (ii) in our experiments $\lambda_{h_u}$ before $\mathrm{d}(h_u,0)$ is kept at a constant value. As for the relationship between the $\lambda$ and $\mu$, they act as dual variables like in a classical Lagrangian dual formulation of a constrained optimization problem. Typically $\mu$ is a decreasing function of $\lambda$. In the linear case of section 3.3, one can show that the primal and dual formulation are equivalent [1].
> We added a comment in the main text to give an intuition about the hyperparameters.
>
> ## About Section 3.3
> We understand your recommendation,yet we believe that for the theoretically oriented reader this strengthens our message. We now highlight in the main text that this case study is a first step, acting as a sanity check. Indeed, if the convergence is not guaranteed in such a simple case, it would prevent us from moving on to more complex learning settings. We highlight that [2, 3] did not provide a convergence analysis of the proposed algorithm even in this restricted setting. Finally, to get insights on the behaviour of Algorithm 1 when $h_u$ is a deep network is hard and left for future analysis. Nonetheless, section 4 experimentally illustrates the convergence of the hybrid model towards satisfying solutions on various datasets of increasing complexity.
>
> [1]: Lecture notes on ridge regression, _Wessel N. van Wieringen_, 2020
>
> [2] Augmenting Physical Models with Deep Networks for Complex Dynamics Forecasting. _Vincent Le Guen, Yuan Yin, Jérémie Donà, Ibrahim Ayed, Emmanuel de Bézenac, Nicolas Thome, Patrick Gallinari_, ICLR 2021
>
> [3] Neural Dynamical Systems: Balancing Structure and Flexibility in Physical Prediction. _Viraj Mehta, Ian Char, Willie Neiswanger, Youngseog Chung, Andrew Oakleigh Nelson, Mark D Boyer, Egemen Kolemen, Jeff Schneider_, arXiv (2021)

---

> > ### Author Response · Authors · 2021-11-16
> > **Answer to R.TpQa (2/2)**
> >
> >
> > ## Additional Points
> >
> > - We detail in the beginning of section 4. the meaning of Ours eq 4. and Ours eq.5
> > - We have changed the format of the images to avoid the blurry results when zooming in.
> > - The difference between true and estimated will be included in the figures in the appendices.
> > - We now include the predictive results for Neural ODE for PDL and LV.
> >
> >
> > ## Novel Experimental Results
> > As requested, we trained  NODE on the damped pendulum and Lotka-Volterra data. Here are the results.
> >
> > | Model |   |Pendulum|     |Lotka Volterra|
> > | :--------| |: -------------- | |: --------------|
> > | | | Pred logMSE| | Pred logMSE|
> > | Ours eq.4 |   |-13.7 (0.84)|  |-9.28 (0.75)|
> > | NODE|  |-10.1 (0.32)| |- 9.1 (1.1)|
> >
> > Interestingly, Our proposition provides better prediction results on the pendulum data whereas the gap with LV is tighter. This may be due to the fact our parameter estimation is more accurate in the case of the Pendulum rather than on LV data.
> > However, note that physics agnostic NODE algorithms cannot be interpreted, i.e. they do not provide direct estimates for the period of the pendulum, or for the growth (resp. extinction) rate of preys (resp. predators).

---

### Official Review · Reviewer_YsB3 · 2021-11-04

**Correctness:** 3
**Technical Novelty And Significance:** 3
**Empirical Novelty And Significance:** 3
**Recommendation:** 6
**Confidence:** 3

**Main Review:**

The paper is generally clear and well-written and the proposed approach is interesting and well supported by experiments.

With that said, I have a few comments:
Theory:
* Proposition 1: It is not clear how the uniqueness of the solution is connected to the accurate identification of h_k. My main concern here is that the paper does not place any condition on the relation between the function spaces H_k and H_u. Contrast this to the linear case where A is invertible and thus cannot realize a constant derivative without D_A.

* It appears that Section 3.3 abandons the "realizable setting" assumption, where the true function is assumed to be in the hypothesis space (which is good). I suggest that the authors put more emphasis on this. Otherwise, the transition can be confusing.

* Proposition 2: I am not sure of the implication of the proposition. It seems to only suggest that for each \hat{A} there exists a unique D_A but does not suggest the uniqueness of A, which is what we would be interested in for interpretability.

Experiments:
A question that arises is whether alternating optimization is needed in practice or if joint gradient descent will be as good. It would be interesting to see this experiment.

Clarity:
* Section 4.2 needs much larification:
Equation 17: The del operator needs to be defined. What does TU mean when U is a tuple (u, v). Why do you use parenthesis for S(U) but not TU. Also, that does T_{t-k} mean? Doesn't k stand for "known"?
The fact that \theta^* is a function in time (rather than a constant vector) can be confusing and should be spelled-out.

* After eq (16) "\theta^* = (\alpha, \beta)", I believe \beta should be \gamma?





**Summary Of The Paper:**

The paper proposes a method for hybrid model-based/machine-leanring learning, wherein a model is decomposed into an interpretable parametric prior and a residual (typically represented by a neural net). The paper demonstrates that prediction error minimization does not accurately identify the parametric component and proposes an alternating optimization method that augments prediction error loss with component-specific losses. The paper also introduces a variant where auxiliary data can be used to define the loss.

Theoretical guarantees are provided where auxiliary losses are given as upper-bound constraints and exact minimization is performed. Also, uniqueness and convergence guarantees are provided for the linear approximation case using a more practical algorithm.

Experiments show the efficacy of the method in linear and non-linear settings, in both identification and prediction.

**Summary Of The Review:**

The paper proposes an interesting method to train hybrid models for both identification and prediction quality.
Connected two ideas from the literature in a single framework is also a welcome contribution. The proposed method is supported by experiments on different domains. Theoretical aspects are promising but still limited.

---

> ### Author Response · Authors · 2021-11-16
> **Answer to R.YsB3**
>
>
> Thank you very much for the positive review.
>
> ## Questions on the theoretical part
>
> ### On Proposition 1, Uniqueness and Identification
> - You are right. Proposition 1 uniqueness result provides a necessary condition for identification. It does not require additional assumptions on the functional spaces, besides relative compactness of S_k, to recover uniqueness. But this does not formally ensure identification. The identifiability is “demonstrated” empirically: for all the all tested settings, the proposed constraints indeed enable accurate identification.
> - On proposition 1 and 3: Both provide uniqueness results, but proposition 3 requires additional assumptions. In the case of proposition 1, the uniqueness is recovered using distance $\mathrm{d}$ between the considered functions. This distance is not computable in practice. We then chose to rely on the distance between induced trajectories, i.e. between ODE flows: $\mathrm{d}_\phi$, as described in section 3.2.2. Thus in section 3.3 the goal is to work on the distance  $\mathrm{d}_\phi$. This analysis is more challenging and requires additional assumptions.
>
> ### On Section 3.3
> As duly highlighted, the true $f_k$ may be a non linear function, and then $h_k$ which is considered linear in this section does not belong to the same space.  We have added a sentence detailing this in the main text of the updated version.
>
> ### On the uniqueness of $A$
> The goal of section 3.3 is to recover the uniqueness in the estimation of both $A$ and $D$. A theoretical case for the well-posedness in $D$ is considered in Proposition 2, relying on an orthogonal projection lemma. The case for $A$ is more complex. As discussed in section 3.3, the uniqueness of $A$ by minimizing eq. (13) is not guaranteed. We then consider discretizing the trajectories of $X_t$. This leads to a least square estimation for $A$, for which the well-posedness is well known so that this estimator is unique. We clarified in section 3.3 that discretizing the trajectories and estimating $A$ via least-square estimation enables us to recover the well-posedness.
>
> ## Question on the experiments
> The comparison between the alternate and joint optimization is already investigated in table (3) and commented in the experimental section 4.2 (paragraph Ablation Study). This shows that alternate optimization performs significantly better than joint optimization.
>
> ## Clarity
>
> ### About the geophysical data
> - $\nabla . (TU)$ refers to the advection of a scalar quantity $T$ by a velocity field $U=(u,v)$ and writes as : $\nabla.(TU)= \frac{\partial T}{\partial x} u +  \frac{\partial T}{\partial y} v$. We clarify this notation in appendix E.3.
> We write $S(U)$ to highlight that the source term $S$ depends on the unknown velocity field $U$. In the case of the simulated data, the formula for $S$ is given in appendices E.3 and in the Natl simulation the true $S$ is unavailable and corresponds to a combination of several very different physical processes e.g. ocean-atmosphere heat exchanges, interactions with deep ocean layers, and so on.
>
> - $T_{t-k}$ refers to $T$ observed at time $t-k$, this notation is indeed unfortunate and we replace it with $T_{t-l}$, where $l+1$ is the length of the observation history of the temperature $T$ required for the estimation of $U$.
>
> - The fact that $U$ varies with time is explicit in the generic dataset description (just after eq.17). We will emphasize that $\theta_k$ varies with time in the experimental setting.
>
> ### About LV
> Indeed, $(\alpha, \beta)$ should be $(\alpha, \gamma)$. This has been updated in the novel version.

---

### Author Response · Authors · 2021-11-16
**Overall Response 1/2**

We would like to thank the reviewers for their feedbacks and look forward to discussing the raised questions.
We individually responded to each review and updated our submission to address the concerns (additions are in blue in the re-submission).
Notably, we incorporate the additional experimental results for NODE as requested by reviewer TpQa.

Let us remind the main contributions of the work:
- We introduce a novel way to recover well-posedness in the learning of hybrid  NN / Physics models through the control of an upper bound
- We further extend our framework to incorporate auxiliary data when available to handle complex real-world problems.
- We propose a novel alternate-optimization algorithm to learn hybrid models, verify its convergence in a simplified case and experimentally evidence the soundness of our approach on settings of increasing difficulty, including challenging real world problems.

---

> ### Author Response · Authors · 2021-11-22
> **Overall Response 2/2**
>
> As a final addition to the rebuttal version, we added in the appendices 2 images including differences in the estimation in the various scalar fields for AdV+S data as required by R. TpQa.
>
> We hope it clarifies our experimental section and look forward to discussing any comments or questions.

---

### Decision · Program_Chairs · 2022-01-20

**Decision:**

Accept (Poster)

**Comment:**

The paper proposes a method for hybrid model-based/ML learning, where a model is decomposed into an interpretable parametric prior and a neural net residual.  In this case, the prediction error minimization does no identify the parametric component, and an alternating optimization method is proposed to augments prediction error loss with component-specific losses. Empirical and theoretical results are obtained.  Initial questions of several reviewers were addressed.